# OSM+: Cloud-native Open Street Map Data System for City-wide Experiments

## Abstract

Road network data can provide rich information about cities and thus become the base for various urban research. However, processing large-volume world-wide road network data requires intensive computing resources and the processed results might be different to be unified for benchmark downstream tasks. Therefore, in this paper, we process the OpenStreetMap data and release a structured world-wide 1-billion-node road network graph database with high accessibility and usability. We have presented three illustrative use cases, traffic prediction task, city boundary detection task and traffic policy control task. Moreover, for the well-investigated traffic prediction task, we release a new benchmark with 31 datasets, which is much more comprehensive than the previously frequently-used datasets. While for the relatively novel traffic policy control task, we release a new 6 city datasets with much larger scale than the previous datasets. Along with the OSM+ dataset, the release of data converters facilitates the integration of multimodal spatial-temporal data based on map information for large model training, thereby expediting the process of uncovering compelling scientific insights.

## 1 Introduction

Road network has formed the skeleton of cities, as it connect between regions within city and between different cities. For long time, urban regions and road networks stretch along each other. Therefore, road networks can essentially reflect the landscape and function zones in cities, and thus affect human mobility. For instance, sky-scrappers, restaurants, and shopping centers tend to locate at places with dense in-city road networks. Hence, investigating road network structures is the base for urban research, e.g., urban planning, urban traffic prediction.

However, obtaining accurate road network data for open public research is difficult for the following two reasons. (1) Accurate road network data is collected at extremely high expense by map providers like Google (Google Maps), Bing (Bing Maps), Baidu (Baidu Maps) and Gaode (Gaode Maps). Thus, these products are mainly designed for commercial use with only very limited specific high-level API open to the public, e.g., POI searching, origin-destination route planning. This can not satisfy the need of academic researchers and start-up companies to conduct flexible low-level computing operations on open road network data to quickly iterate ideas or products. (2) Open-source map services, e.g., OpenStreetMap (Haklay & Weber, 2008a), built from crowdsourcing mechanisms by world-wide users, seems to be the cure. However, due to the massive amount of the road network data and the complex data format in map object storage, processing the road network data from scratch to obtain desired format for experiments are always challenging and time-consuming. Hence, it is highly desirable that an intermediate format of processed road network data can support diverse downstream applications so as to speed up the scientific discovery.

Following this path, some studies (Grinberger et al., 2022; Bartzokas-Tsiompras, 2022; Ding et al., 2022) have been conducted utilizing open road network data like OpenStreetMap, while several issues remain. First, the cleaning of OpenStreetMap contains a complex pipeline, including converting, reducing, transforming and aggregating. This pipeline may take about 10 hours even only for processing a region with $1,000km^2$ size, when running on a computer with 32 CPU cores and 128GB memory. Second, the computing of world-wide map data requires memory far more than that of a single machine. The world-wide raw OSM data is roughly 1.1 TB before processing, and finally processed structured OSM data can be a graph with more than 1 billion nodes. Although segmenting

the regions and process each region separately may make it possible to process the data locally, this requires hand-crafted distributed computing strategies and further aggregating the processed data may introduce extra errors to the data. Because of the aforementioned two issues, researchers created different ways to process the road networks and thus obtained various versions of same raw data. In this large model era, this has become the barrier for comparison among different studies and creation of multi-modal spatial temporal data for large model training. Hence, these issues can be solved with building a pre-computed structured version of OpenStreetMap for public use with a powerful computer.

In this paper, we propose to make public a structured road network computing engine, OSM+. OSM+ is composed of three components: (1) a road network graph database with world-wide intersections (nodes) and road segments (links), and side information contained in map, e.g., POI; (2) a series of auto-parallel fundamental computing APIs to allow efficient node query and distance query, on which researchers may build more comprehensive road network analysis; (3) a series of data converters to tailor the road network data for three typical urban research problems. The contributions of this paper can be summarized as follows.

- **Data-wise**. We provide an easy-to-use world-wide road network database and make it open-source. This makes the dataset a benchmark for related research. The OSM+ dataset can be accessed and the related code can be found at `https://anonymous.4open.science/r/OSM-dataset-3034`.

- **System-wise.** We provide cloud-computing based APIs to enable efficient billion-scale graph query and processing, allowing various subsequent data processing for extended applications.

- **Application-wise.** We provide three example application scenarios, traffic prediction task, traffic signal control task and city boundary detection task. The data converters are released, which will make it possible to use map data as the base to fuse multi-modal spatial temporal data for large model training and accelerate interesting scientific discovery.

- **Benchmark and standardization.** For these three tasks with different investigation levels by the community, we provide different benchmark and standardization achievements. For the well-investigated traffic prediction task, we construct a new large-scale benchmark. with 31 cities. This contains much more comprehensive data than the 6 previously commonly-used datasets (from PEMS and METR-LA) and further covers more challenging in-city scenarios. For traffic policy control task, we release 6 city datasets with each city containing at most 18,948 intersections. This scale is much larger than the datasets used by most of the previous studies. While for the city boundary detection task, we clearly define this novel problem which may attract more people to develop new methods in this field. These data will largely push the three fields forward on city-scale modeling and generalizability to various cities.

## 2 RELATED WORKS

**Map databases and services.** Map databases and services are the base for various research. The major map databases include two types, commercial services and open-source maps. The commercial services include Google (Google Maps), Bing (Bing Maps), Apple (Apple Maps), Baidu (Baidu Maps), and Gaode (Gaode Maps). Though varying in serving regions, these map providers usually only provide high-level APIs (e.g., plotting, routing) for users to develop their own applications. This kind of APIs can not support research need for querying external elements on the map, e.g., population count. To facilitate faster research progress and small business, some open-source map providers release their products, including OpenStreetMap (Haklay & Weber, 2008b; Bennett, 2010; Mooney et al., 2017), Mapbox (Eriksson & Rydkvist, 2015), Leaflet (Edler & Vetter, 2019), GeoServer (Kshetri et al., 2021). These products store the map in various formats, which are difficult to be processed into a uniform format and need to be further cleaned in order to research use. Hence, in this paper, we propose to solve this problem by mapping the open-source map information into a well-structured graph database, with road network as the bone, and other information as the attributes. This intermediate format will accelerate numerous downstream applications.

**Map computing engine.** To better utilize map data, many map computing engines have been invented. These tools can be categorized into two types: commercial map tools and open-source map tools. For *commercial map engines*, Hu & Dai (2013) developed an online map application based on the Google

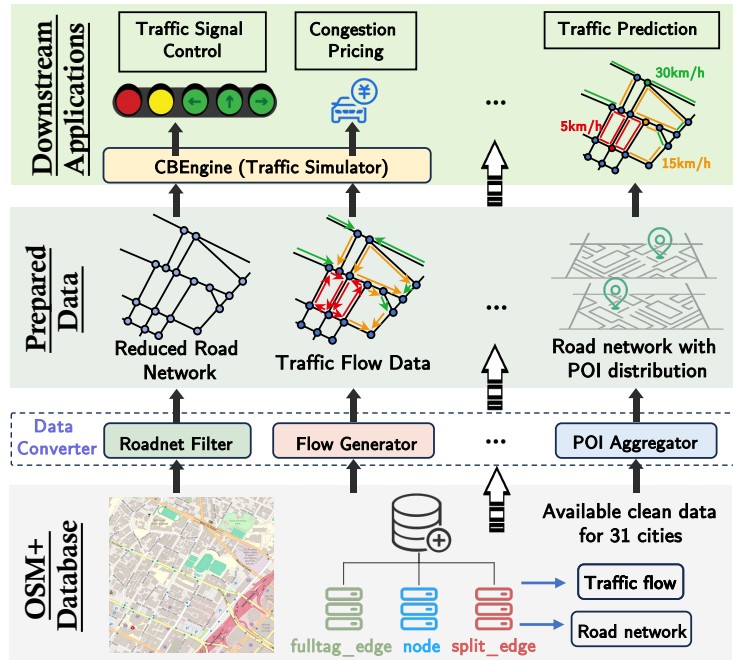

Figure 1: The overall framework of OSM+.

Maps API, using commercial databases to provide users with complex data manipulation functions. Amani et al. (2020) processed remote sensing map data of Canadian agriculture on the Google Earth Engine (GEE) and obtained the annual crop list of Canada. Nguyen et al. (2019) combined GEE with the automatic water extraction index (AWEI) to address the long processing times in monitoring water surface dynamics. For *open-source map engines*, Elleuch et al. (2014) accomplished the process of converting large-scale databases collected by cars into road tracks. Boeing (2017a;b) developed OSMnx, which simplifies data collection and road network analysis from the perspectives of graph theory, transportation, and urban design. Although these tools succeed to help process the map data efficiently, they still have two deficiencies. First, the processed data are not well-defined uniform structured data and are usually only used for one-time use. Thus, repetitive computation are needed when the map data is needed for another similar use. Second, the data processing program is running on local machines, which limits the capability to extend to large-scale analysis, e.g., world-wide.

## 3 OSM+: READY-TO-USE WORLD-WIDE ROAD NETWORK DATABASE

To solve the aforementioned problems, we propose OSM+ (OpenStreetMap Plus), a structured road network computing engine. It is composed of three components: (1) a road network graph database that includes world-wide intersections (nodes) and road segments (links), and supplementary information from maps, e.g., POI; (2) a series of auto-parallel fundamental computing APIs to allow efficient node query and distance query; (3) a series of data converters designed to customize the road network data for typical urban research problems.

### 3.1 GRAPH DATABASE

We generalize the road network as a graph with 1.9 billion nodes under the following definition.

- Node: Each node represents a road intersection in the OpenStreetMap road network or POI point.

- Link: Each link represents a road segment in OpenStreetMap with a starting intersection and ending intersection. Since each road segment may have multiple parts segmented by minor intersections or direction turning point in geometry, each link may contain multiple line segments.

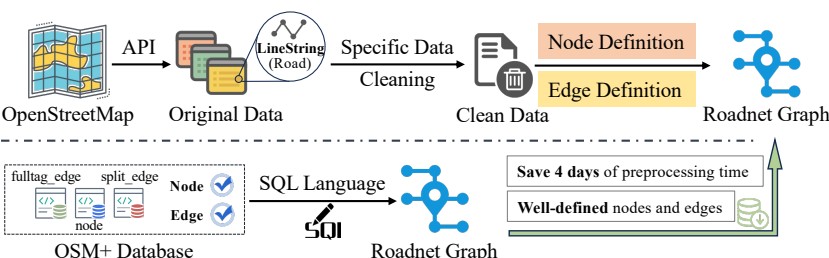

Figure 2: Graph database with worldwide edges and points by OSM+.

Based on these definitions, we may project urban concepts into graph concepts. Note that, a lot of downstream machine learning tasks in the transportation domain, such as traffic prediction, rely on road network graph structures. Our OSM+ dataset, which is a graph-structured database based on ODPS, can thus offer significant advantages for researchers. It facilitates quick queries and seamless table joining, thereby enhancing the efficiency of traffic prediction tasks and other studies when you need to directly use the world map data as a graph, or need to join the map data with another dataset.

**Overall numbers.** The overall statistics are shown in Table 1. Since in the original OpenStreetMap, road is represented by a linestrings, we divide linestrings into several "split edges" with only one start point and one end point. The "split edges" representing the road are called "split edges on roadnet". It is easy to observe that the worldwide road network is a huge graph with billions of nodes and edges. In addition, the density of points may suggest the different population and civilization levels.

**Spatial distribution & categories.** It is obvious that the spatial distribution of road data is imbalanced. Thus, the number of road network nodes and the number of road network edges in terms of each continent are further shown in Figure 3 (a). Europe has the most, Asia is the second, and Oceania has the least. Further, we calculate the total total length of the road network in the five countries or regions, and we find the International Road Federation statistics for the total road length in the five countries or regions in 2019. These statistics are presented in Table 2. Comparing these data, we can find that the data of OSM in the United States, the Russian Federation and Canada are more detailed, while the data in China and India are relatively incomplete. The category statistics is shown as in Figure 3(b). It is observed that the number of roads increases exponentially as the road level drops.

Table 1: Some basic statistics of our OSM+ database.

| Roads and POI | | | Roadnet Only | | | Total |
|---|---|---|---|---|---|---|
| **Nodes** | **Split Edges** | **Linestrings** | **Nodes** | **Split Edges** | **Linestrings** | **Length(km)** |
| 7,475,535,808 | 1,964,857,157 | 8,039,543,656 | 2,180,447,343 | 833,401,275 | 197,775,476 | 84,662,999 |

Table 2: Total length of road (km) in different countries from OSM+ and IRF data sources.

| Data Source | China | America | Russia | India | Canada |
|---|---|---|---|---|---|
| **OSM+** | 3,805,919 | 13,731,271 | 9,956,729 | 2,878,553 | 1,592,642 |
| **IRF Statistics** | 5,012,496 | 6,638,329 | 1,542,196 | 6,371,847 | 1,126,600 |

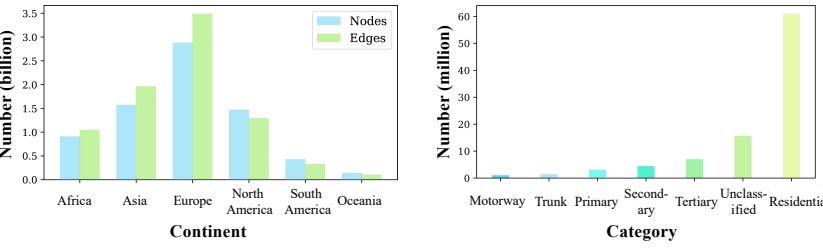

Figure 3: Basic continent and category statistics of OSM database.

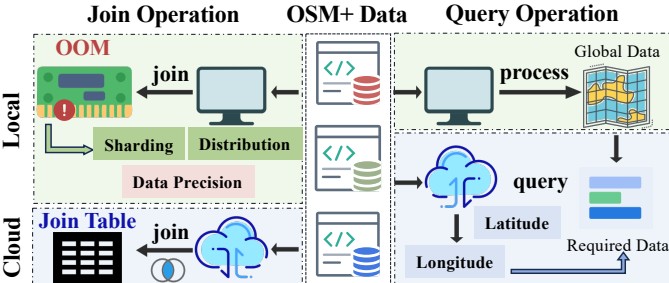

Figure 4: OSM+ (on cloud) provides efficient and easy-to-use Query APIs. If global data is processed on a local device (w/o OSM+), out of memory may be caused by the join operation due to the large data table (top left). In addition to this, it is cumbersome to process all the data in the region when constructing a specific region map data (top right).

## 3.2 EFFICIENT QUERY APIS

We introduce how to use cloud computing resources to enable efficient parallel query of the OSM+ data. In comparison to directly downloading data from the OpenStreetMap API, using the OSM+ dataset can save approximately 4 days of preprocessing and cleaning time for global data. Meanwhile, we implement several simple optimization techniques, "window check" and "k-d tree" as examples. Researchers are welcome to implement their own optimization tricks.

**Point Query Operation** Point query operation is one of the most basic query operations. Given a point and a radius, select all points within a given radius around the given point in osm_node. A simple method is to traverse all other points in the dataset and filter the points that meet the conditions. However, doing so will result in a huge amount of calculations. "window check" technique first divides the geographic space evenly into several grids according to a certain accuracy, and then only needs to retrieve data points in adjacent grids during retrieval. We conduct 1,000 point queries with/without "window check" on cloud computing infrastructure. The results in Table 3 show that "window check" can significantly improve the efficiency of point query.

**Nearest pair query and k-Nearest pair query** Given a point and a latitude and longitude range, the aim is to find the nearest point or points in this latitude and longitude range. We apply k-d tree, a binary tree that represents a division of $k$-dimensional space to reduce the time complexity. Constructing a k-d tree from OSM+ dataset is equivalent to continuously dividing the $k$-dimensional space with hyperplanes perpendicular to the coordinate axis to form a series of $k$-dimensional hyperrectangular regions. Since OSM+ dataset has huge amounts of records but low dimension (which means $n >> d$), the optimal time complexity of k-d tree to find the nearest neighbor is $O(\log_2 n)$. As shown in Table 3, the k-d tree technique can significantly reduce the running time needed for this query.

Table 3: Effectiveness of adding "window check" and "k-d tree" optimization technique.

| Method | Point Query | | Nearest Pair Query | |
|---|---|---|---|---|
| | Runtime(s) | Core $\times$ min | Runtime(s) | Core $\times$ min |
| **With Optimization** | 79 | 1.33 | 1.49 | 0.01 |
| **W/O Optimization** | 310 | 2.06 | 41.14 | 0.05 |

**An Example on Using Optimized Query** Based on these basic APIs, we build an example comprehensive calculation task to conduct the KDE kernel density estimate of each intersection node on the global road network data, to illustrate why it is necessary to run experiments on cloud computing. We use three different sampling rates to sample the original global road network data and compare the runtime and memory cost by different platforms. The experimental results are shown in Table 4. It is observed that by utilizing the ODPS computing engine, we could employ optimized query algorithm which significantly outperforms that of other computing platforms in terms of both runtime and memory utilization. Moreover, its judicious use of memory resources minimizes the memory

Table 4: Efficiency of global KDE estimation on different computing cluster.

| Sample Rate Platform | 1/1,000 | | 1/10,000 | | 1/100,000 | |
|---|---|---|---|---|---|---|
| | Runtime(s) | Core × min | Runtime(s) | Core × min | Runtime(s) | Core × min |
| ECS | OOM | OOM | 841.62 | 14.02 | 6.04 | 0.10 |
| Spark | 29,014.66 | 17,408.49 | 197.02 | 118.21 | 7.21 | 4.32 |
| ODPS | 72.13 | 1.02 | 34.26 | 0.30 | 33.06 | 0.22 |

cost, enabling query tasks on large-scale datasets. Note that, continuing to increase the sample rate (e.g., 1/100, 1/10) will make the other two platforms fail to finish, hence, these results are not reported. Consequently, ODPS not only enhances computational speed but also mitigates memory resources cost, making it a good choice for applications on OSM+ dataset.

## 4 TYPICAL APPLICATION SCENARIOS

In this section, we introduce three typical applications of OSM+ database. To establish a benchmark at the city scale, one needs a comprehensive roadnet graph, and compile their dataset with the roadnet data. This initiative could be easily built upon our provided OSM+ dataset and enables researchers to curate a dataset specifically for these benchmarks.

### 4.1 CITY-SCALE TRAFFIC PREDICTION AND TRAFFIC GENERATION

Traffic prediction has been investigated a lot by researchers due to its important applications. Studies usually use the famous PEMS datasets and METR-LA dataset for experiments. However, in recent years, it has been noticed that newly proposed methods can hardly exhibit significant improvements over existing ones. Thus, we are here to propose **31 new city-level datasets** associated with traffic flow data, to provide a more comprehensive benchmark for this problem. In addition to the many more cities compared with previous datasets, the newly proposed datasets are different from previously used datasets for several reasons as shown in Figure 5.

- **Dynamic in-city Scenario:** Unlike previous datasets, which may have focused on more uniform highway conditions, these datasets capture a broader spectrum of in-city scenarios. This introduces greater variability in the data, reflecting the diverse and dynamic nature of urban environments. Such variability is crucial for developing models that can effectively handle the complexities and unpredictability inherent in city traffic patterns and infrastructure.

- **Sparsity Challenge:** The number of sensors is relatively low compared to the road intersections of segments. Modeling sparsity is critical as it mirrors real-world conditions where data points can be irregular or missing. Addressing sparsity effectively can significantly improve the accuracy and reliability of the model, ensuring it performs well even in less-than-ideal data conditions. This aspect of the dataset pushes the boundaries of current modeling techniques, encouraging the development of more sophisticated and resilient algorithms.

To evaluate the performance of OSM+ datasets, we test 7 frequently-cited baseline methods on these 31 cities. We test these algorithms on three different prediction horizons (3,6,12) following the widely used setting. Due to the limited space, the average results (over different prediction horizons) are shown in Table 5. It is easy to observe that these methods perform completely differently on these datasets. Compared with the benchmark results in previous papers (Shao et al., 2023), we can have the following conclusions. (1) New datasets induce more variance and bring a more challenging problem for these methods to work on. (2) Currently, there is no single dominating method that can outperform other methods on most of the datasets. Therefore, these datasets will bring great stimulation for the development of this field.

Table 5: Experiment results of traffic prediction task on 31 cities in OSM+(UTD19) with 7 baseline methods. Two metrics MAE and MAPE(%) are reported. Best results in each row are in bold.

| City | AGCRN | | Crossformer | | DCRNN | | DLinear | | FEDformer | | GWNet | | MTGNN | |
|------|-----|------|-----|------|-----|------|-----|------|-----|------|-----|------|-----|------|
| | MAE | MAPE | MAE | MAPE | MAE | MAPE | MAE | MAPE | MAE | MAPE | MAE | MAPE | MAE | MAPE |
| AA | 47.92 | 44.03 | **44.40** | **35.34** | OOM | OOM | 47.80 | 37.84 | 51.94 | 45.19 | 47.06 | 37.14 | 46.97 | 40.47 |
| BSL | 64.51 | 55.38 | 62.82 | 65.68 | 119.15 | 184.45 | 61.50 | **51.74** | **59.37** | 61.78 | 81.07 | 106.88 | 78.41 | 93.81 |
| BRN | 51.00 | 231.93 | **49.84** | **201.84** | OOM | OOM | 52.18 | 253.27 | 55.58 | 248.00 | 50.59 | 319.09 | 70.90 | 405.07 |
| BHX | 112.08 | 70.94 | **84.13** | **48.08** | 303.46 | 195.15 | 111.22 | 66.91 | 119.52 | 65.15 | 107.09 | 66.83 | 91.68 | 49.44 |
| BOL | **31.27** | 21.07 | 32.74 | **20.98** | 38.00 | 26.82 | 37.27 | 34.12 | 37.87 | 29.83 | 35.03 | 25.67 | 32.31 | 21.12 |
| BOD | 71.65 | 39.69 | 67.13 | **36.19** | 232.07 | 276.51 | **67.13** | 44.54 | 70.14 | 46.29 | 74.18 | 57.18 | 89.14 | 56.70 |
| BRE | **56.31** | 36.52 | 58.08 | **34.22** | OOM | OOM | 63.27 | 42.47 | 61.42 | 41.98 | 57.01 | 36.98 | 56.69 | 35.50 |
| KN | OOM | OOM | 44.78 | **48.85** | 117.40 | 292.38 | **38.69** | 61.19 | 40.98 | 67.71 | 43.85 | 55.18 | 47.89 | 75.01 |
| DA | 57.22 | 51.76 | **53.28** | 50.75 | OOM | OOM | 54.76 | 53.99 | 57.41 | 61.16 | 54.69 | 51.74 | 57.20 | **50.30** |
| ESS | 41.95 | **34.65** | 40.47 | 41.68 | 174.64 | 294.77 | 50.35 | 43.85 | 44.70 | 41.95 | 38.99 | 34.87 | **38.41** | 36.46 |
| FRA | 163.16 | 54.95 | 145.88 | 47.61 | 179.03 | 51.37 | **99.62** | **30.19** | 107.78 | 31.76 | 190.85 | 62.23 | 284.10 | 92.89 |
| GRZ | 61.15 | 113.62 | **52.78** | 66.74 | 183.88 | 464.62 | 60.83 | **72.86** | 56.16 | 73.15 | 58.03 | 68.32 | 56.60 | 74.38 |
| GRQ | 69.64 | 35.57 | 68.02 | **32.53** | 158.26 | 114.53 | **66.03** | 37.54 | 79.09 | 42.20 | 67.99 | 34.95 | 74.99 | 39.00 |
| HAM | 46.50 | 44.89 | 44.49 | 44.87 | 97.85 | 108.12 | 46.69 | 49.81 | 47.85 | 50.69 | **44.25** | **43.83** | 45.02 | 44.18 |
| INN | 72.80 | 31.56 | 69.28 | 37.40 | 342.05 | 314.50 | 89.95 | 39.55 | 74.44 | 32.32 | **67.03** | **28.53** | OOM | OOM |
| KS | 81.26 | 106.06 | 86.38 | 118.43 | 233.90 | 427.98 | 75.29 | 107.43 | 89.83 | 127.22 | **71.23** | **94.88** | 191.45 | 316.68 |
| MAN | 106.16 | 42.54 | 97.48 | 41.10 | 336.42 | 280.35 | 101.38 | 46.21 | 110.81 | 52.15 | **95.91** | **38.95** | 97.30 | 40.74 |
| MEL | 50.24 | 45.88 | **45.36** | 42.73 | OOM | OOM | 63.72 | 66.55 | 53.25 | 56.25 | 51.91 | **36.10** | 45.48 | 40.26 |
| RTM | **52.48** | **40.29** | 53.83 | 50.52 | 179.76 | 347.68 | 68.83 | 53.19 | 68.43 | 65.17 | 67.03 | 50.91 | 57.34 | 41.07 |
| SDR | 103.63 | 59.74 | 102.25 | 65.34 | 262.60 | 271.61 | 97.97 | 54.70 | 125.51 | 95.71 | **89.36** | 47.38 | 97.54 | **44.61** |
| SP | 49.08 | 39.57 | **47.93** | 37.74 | 119.56 | 119.22 | 52.95 | 45.39 | 53.42 | 44.88 | 48.34 | 38.34 | 48.05 | **37.48** |
| SXB | 78.34 | 39.40 | 76.17 | 38.72 | 261.11 | 223.11 | 85.62 | 46.72 | 84.71 | 46.10 | 76.86 | 39.46 | **76.01** | 37.36 |
| STR | 58.93 | 20.37 | 56.60 | 19.52 | 68.19 | 23.30 | 65.80 | 24.52 | 68.38 | 23.48 | **55.80** | **19.05** | OOM | OOM |
| TPE | 136.50 | 48.04 | 134.51 | 48.18 | 502.95 | 274.25 | 142.61 | 46.21 | 149.12 | 53.31 | **129.13** | **40.14** | 130.36 | 41.42 |
| TO | 89.48 | 57.66 | **81.70** | **44.44** | 314.62 | 390.29 | 85.13 | 48.01 | 87.85 | 56.18 | 102.69 | 60.64 | 104.28 | 68.82 |
| YTO | 51.73 | 39.35 | **51.54** | 40.18 | 161.46 | 145.72 | 90.53 | 71.76 | 62.92 | 59.10 | 58.04 | 38.73 | 52.24 | **37.42** |
| TLS | 257.82 | 751.49 | 255.29 | 756.09 | 268.70 | 847.32 | 263.95 | 870.21 | 296.55 | 836.03 | **255.26** | 751.62 | 258.62 | **730.09** |
| UTC | OOM | OOM | 50.35 | 62.80 | OOM | OOM | 50.78 | 54.42 | 66.80 | 88.25 | 74.98 | 88.33 | **39.92** | **36.74** |
| VNO | 88.95 | 54.81 | 84.09 | 49.34 | OOM | OOM | 76.03 | 43.69 | 88.84 | 49.53 | **73.80** | **39.27** | 96.47 | 64.87 |
| WOB | 54.48 | 41.34 | **52.21** | **39.71** | 0.44 | 47.61 | 62.24 | 50.94 | 57.60 | 50.15 | 54.32 | 42.30 | 53.24 | 40.17 |
| ZRH | OOM | OOM | 54.73 | 36.93 | OOM | OOM | 60.36 | 43.84 | 60.12 | 43.74 | 66.51 | 53.31 | **53.52** | **35.16** |
| **# Win** | 3 | 2 | 10 | 10 | 0 | 0 | 4 | 2 | 1 | 0 | 9 | 8 | 4 | 8 |

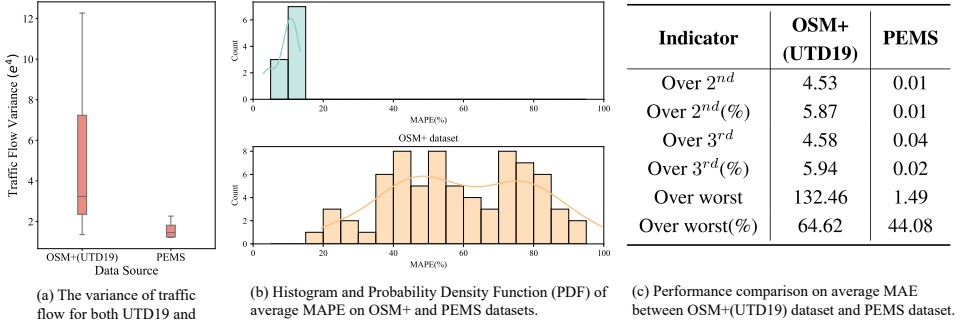

(a) The variance of traffic flow for both UTD19 and PEMS dataset.

(b) Histogram and Probability Density Function (PDF) of average MAPE on OSM+ and PEMS datasets.

(c) Performance comparison on average MAE between OSM+(UTD19) dataset and PEMS dataset.

Figure 5: Comparison between OSM+(UTD19) dataset and PEMS dataset in three aspects.

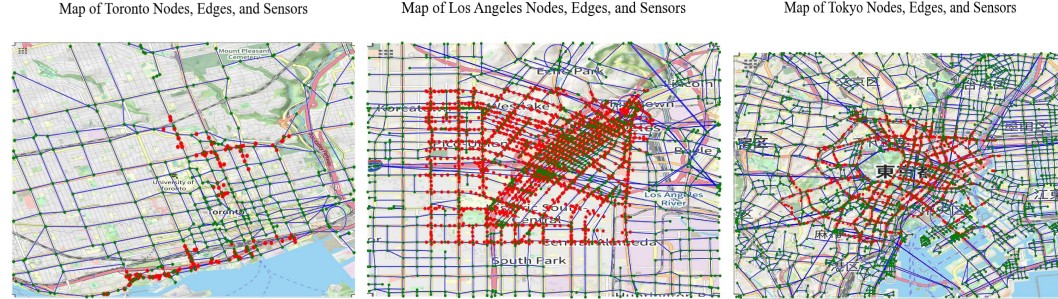

Figure 6: Map of Toronto, Los Angeles and Tokyo. The green points on the graph represent road nodes in OSM+, the blue lines represent road segments in OSM+, and the red points represent loop sensors. It can be seen that the road network information of OSM+ is very consistent with the actual ground truth. The loop sensors cover almost all the nodes in the central area of the city, which can better reflect the traffic flow information of the city.

Table 6: Basic statistics of traffic signal control experiments.

| Indicator | Former datasets | | OSM+ dataset | | | | | |
|---|---|---|---|---|---|---|---|---|
| | Hangzhou | Manhattan | New York | Los Angeles | Beijing | Shanghai | London | Paris |
| Intersections | 16 | 2,510 | 5,971 | 6,663 | 18,948 | 14,750 | 5,895 | 1,721 |
| Vehicles | 2,983 | 48,079 | 90,059 | 112,291 | 130,851 | 85,480 | 107,105 | 101,929 |

### 4.2 TRAFFIC POLICY CONTROL

Based on data from OpenStreetMap, we can further build simulation environments for traffic policy control experiments (e.g., traffic signal control). The typical procedure of building the traffic policy experiments is in Figure 1.

Compared with previously-used datasets for traffic signal control experiments (Wei et al., 2019), we can now easily provide benchmark scenarios with a much larger scale. Here, we show six example city scenarios that we have cleaned up. Basic statistics are shown in Table 6. These datasets will allow researchers to work on close-to-reality city-scale traffic policy experiments, which bring both new challenges and opportunities for continuous model improvement.

We have successfully applied this pipeline in the scenario of a real-world city and help tested hundreds of traffic signal control algorithms in this scenario. The best algorithm can improve over 30% compared with the baseline algorithm.

### 4.3 CITY BOUNDARY DETECTION

Modern cities come into being and evolved in the last 200 years. By 2050, more than half of the world population will live in cities (Ritchie & Roser, 2018). This is because the city, as a unit for urban service providing, can boost the efficiency and convenience of the daily life of people, e.g., retailing, delivery, ride-hailing. Meanwhile, the planning of city development is usually supported by local government funding. Hence, it is essential to identify the city boundary so as to improve urban service providing. It is notable that, areas of cities are connected via thousands of roads. Therefore, a straightforward intuition is to detect city boundaries by road network density. Different from the registration boundary, this definition can better illustrate the local economic connection between different areas naturally formed by human activities.

Here, we showcase how to use the OSM+ database to define city boundaries and compare the detected boundary with the actual administrative boundary of the city in Database of Global Administrative Areas (GADM, 2015). By overlapping the two kinds of boundary data, we aim to find out the consistent and contradictory parts between them. If we find that the cluster boundary is not consistent

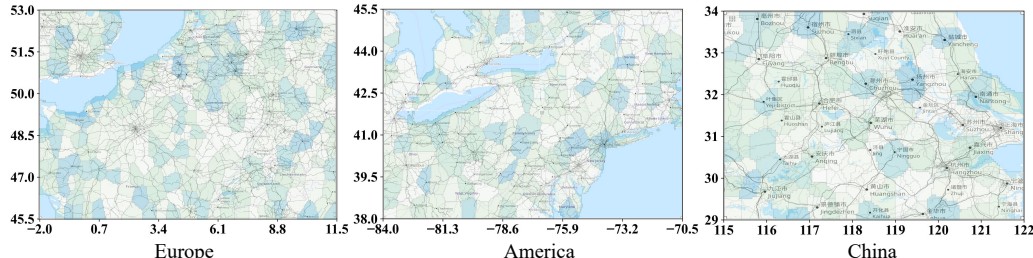

Figure 7: The city boundary map obtained from the clustering results of central Europe, the east coast of the United States and the Yangtze River Delta in China is overlaid with the road maps of the three regions on OpenStreetMap, and latitude and longitude are added to distinguish them.

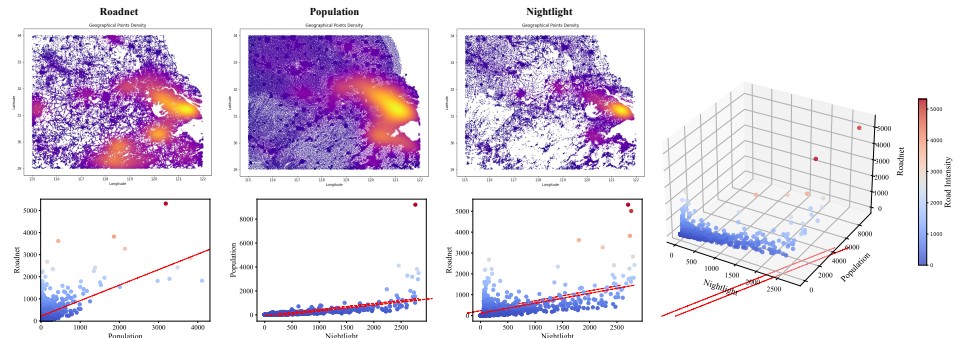

Figure 8: Comparing the road, population and night lighting data map in the same area, the road density is positively correlated with the population and night lighting data density.

with the actual administrative boundary, it may indicate that the city is developing rapidly, showing a trend of gradually expanding its scale.

We overlay the cluster boundary and the actual administrative boundary of the Yangtze River Delta in China, and the result is shown in Figure 7. The different color blocks stand for the obtained clusters while the lines represent the registration boundary. It is easy to observe that some administrative boundaries are made up of several colors, which shows that the areas within a single administrative boundary might be composed of several relatively separated urban areas. The clustering results of many cities match the actual administrative boundaries well, such as Shanghai and Jiaxing in China. The detected borderlines and ground truth of these cities almost coincide. In contrast, there are also some mismatches. This is mostly because there are many roads near the registration boundary of two cities (indicating close economic interactions) and this leads to these two cities being clustered together. This shows the trend of urban agglomeration, and proves the rationality of clustering.

In addition to the direct research on road networks, scholars can also combine road network with other economical data to investigate how road network contribute to the city development and civilization. The 3D scatter plots of the road network, population and night light data of the city are shown in Figure 8. The density of points around a point is represented by different colors in the last two maps. The darker red place in the population data map represents the denser population, and the brighter place in the city night lighting data map represents the more lights at night.

Comparing these three figures, we can easily observe that the dense road network usually indicates dense population and and intensive night lights, i.e., there is a positive correlation among them. In fact, the closer to the city center, the more people and buildings there are, so come more lights at night. In addition, looking at the latter two figures separately, for the population data figure, we can filter out those points whose R value is higher than a certain threshold according to the RGB values of the colors of the points, which are the point sets of the city center, and the range they form is the city center. Similarly, for the night lighting data, we can filter according to the brightness and also get the range of the city center.

## 5 DATA PROCESSING

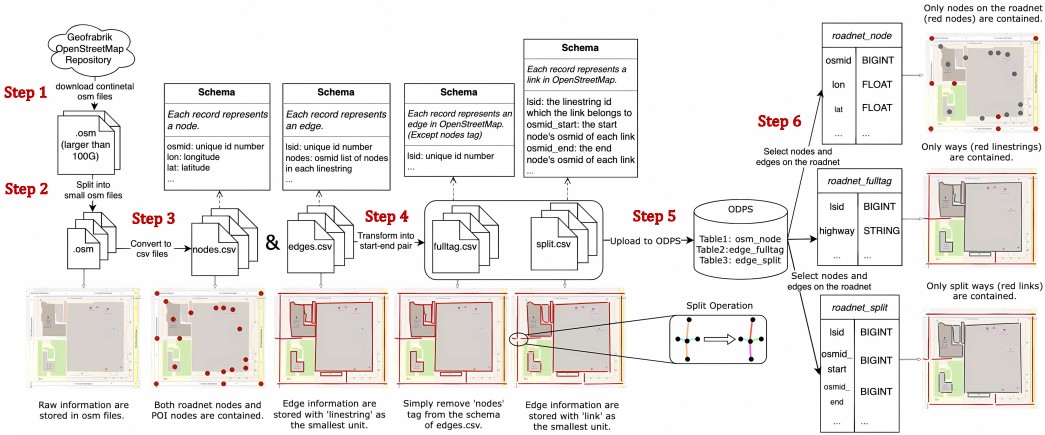

Figure 9: The flow chart of data processing.

The overall procedure of the OpenStreetMap data processing is shown in Figure **9**. It can be described as the following steps:

1. Download continental osm files data from `http://download.geofabrik.de/`.

2. Decompose continental osm files (larger than 100G) into several small osm files.

3. Convert osm files into csv files which can be divided into **nodes.csv** files and **edges.csv** files. In node files, each record represents a node on OpenStreetMap with about 30 attributes. In edge files, similarly, each record represents an edge on OpenStreetMap.

4. Split linestrings on OpenStreetMap and transform **edges.csv** files into start-end pair. Each edge file will be transformed into a **fulltag.csv** file and a **split.csv** file. The schema of fulltag.csv is the same as **edges.csv** except for removing the "nodes" tag. Each record in **split.csv** represents a link on OpenStreetMap. The split operation is shown in Figure **9**.

5. Upload these files to ODPS.

6. Select nodes and edges on the roadnet from tables on ODPS. Finally, we generate three tables on ODPS: *osm_node_roadnet*, *osm_split_edge_roadnet* and *osm_fulltag_edge_roadnet*. The detailed schemas of these three tables are introduced in Section A in Appendix.

## 6 CONCLUSION

In this paper, we introduce a structured road network computing engine called OSM+. OSM+ comprises three main components: (1) a road network graph database featuring global intersections (nodes) and road segments (links), along with supplementary map information such as points of interest (POIs); (2) a series of auto-parallel fundamental computing APIs designed for efficient node and distance queries, providing a foundation for more comprehensive road network analysis; and (3) a collection of data converters to adapt the road network data for three typical urban research problems. We present three example application scenarios: traffic prediction task, and traffic policy control task and city boundary detection. The released data converters allow map data to be used as a foundation for integrating multi-modal spatial-temporal data, supporting large model training and accelerating scientific discovery. Especially, for traffic prediction task and traffic signal control task, we release a new benchmark covering 31 cities and 6 cities respectively. These datasets will significantly advance city-scale modeling and improve the generalizability of research across various urban environments.

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

## A  DATA FIELDS

There are three tables in our dataset, which are: osm_node_roadnet, osm_split_edge_roadnet and osm_fulltag_edge_roadnet. The meanings of the data fields in each table are as follows. Note that, the released data fully describe the road network structure over the world. Meanwhile, OpenStreetMap also contains billions of records for the other types of nodes (including buildings, POIs, etc.) and related edges, 7,475,535,808 nodes and 8,039,543,656 edges in total. Nodes and edges also contain tag information. We have selected 32 node features and 60 edge features which appear the most. We are cleaning these data and processing it to avoid interest conflicts and ethical issues. The release plan for these data and attributes will be determined later.

TABLE **osm_node_roadnet:** (1,964,857,157 rows and 3 columns)

- **osmid** A unique identifier for each node; different nodes or edges have different osmid.
- **x** The longitude of a node.
- **y** The latitude of a node.

TABLE **osm_split_edge_roadnet:** (2,180,447,343 rows and 7 columns)

- **lsid** The identifier of a road segment in the osm_split_edge_roadnet table.
- **osmid_start** The osmid of the starting node of a road segment in the osm_split_edge_roadnet table.
- **osmid_start_x** The longitude of the starting node of a road segment in the osm_split_edge_roadnet table.
- **osmid_start_y** The latitude of the starting node of a road segment in the osm_split_edge_roadnet table.
- **osmid_end** The osmid of the ending node of a road segment in the osm_split_edge_roadnet table.
- **osmid_end_x** The longitude of the ending node of a road segment in the osm_split_edge_roadnet table.
- **osmid_end_y** The latitude of the ending node of a road segment in the osm_split_edge_roadnet table.

TABLE **osm_fulltag_edge_roadnet:** (197,775,476 rows and 2 columns)

- **osmid** A unique identifier for each edge; different nodes or edges have different osmid.
- **highway** The road hierarchy for different roads, starting from the highest level: motorway, trunk, primary, secondary, etc.

If a split edge A in osm_split_edge_roadnet belongs to a fulltag edge B in osm_fulltag_edge_roadnet, then the **lsid** of A is equal to the **osmid** of B. Thus, in table osm_fulltag_edge_roadnet, each edge's **osmid** is unique, with each **osmid** value appearing only once in the table. Conversely, in table osm_split_edge_roadnet, the **lsid** values may be repeated; they are the same if and only if two split edges belong to the same full tag edge.

## B  DATA ACCESS

Our OSM+ dataset is distributed under the CC-BY-SA 4.0 license. Researchers can easily access OSM+ dataset through interfaces provided in `https://anonymous.4open.science/r/OSM-dataset-3034`. Please note that, we only process the dataset from the original OSM dataset rather than create the data. Since OpenStreetMap is a crowdsourcing project, we are not responsible for any potential interest conflicts in the data.

## C  DETAILED EXPERIMENT SETTINGS

### C.1  BASIC QUERY OPERATION

This section describes the details of optimizing basic query operations. For point query, we divide the entire map into grids of 0.2° by 0.2° in latitude and longitude, and determine which grid the point to

be queried is located in. Then, we only need to retrieve data points in adjacent grids during retrieval. For the nearest neighbor query, we also used the "window check" method. In addition, we utilized the k-d tree, a binary tree that represents a division of k-dimensional space to reduce the time complexity.

## C.2 CITY BOUNDARY DETECTION

This section mainly describes the process of using OSM+ data for city boundary detection. First, we select several representative areas for experiment, including China Yangtze River Delta (115°E~122°E, 29°N~32°N), New York in USA (84°W~70.5°W, 37.8°N~45.6°N), Central Europe (2°W~11.5°E, 45.5°N~53°N), Nigeria in Africa (0.5°E~14°E, 4.2°N~11.7°N). We integrate the roadnet data, raster image population data, and raster image nightlight data of these areas, by converting them into point-wise data. Next, we perform clustering on the road net data, nightlight data, and population data respectively, and then weigh and sum the results of clusterings to obtain a new clustering result. After that, we divide the entire map into a grid structure with a size of 0.05° in latitude and longitude. We then count the number of points of each type within each grid and assign the grid to the category with the most points, obtaining a rough boundary. Finally, we examine the empty grids by considering a grid nine times its size centered on it and reclassifying it to make the boundary smoother. This process allows us to discover a more reasonable city boundary.

## C.3 CITY-SCALE TRAFFIC PREDICTION

We conduct all the experiments on machines with two NVIDIA 3090 GPUs and 128 GB memory on Ubuntu 20.04. All models are implemented in Python 3. For the problem setting, we set the input sequence length and output sequence length both to 12. The traffic flow data is extracted from Loder et al. (2020). The ratio of training set, validation set, and test set is 6:2:2 for all 31 city datasets. The evaluation metrics we choose include mean absolute error (MAE), root mean squared error (RMSE), and mean absolute percentage error (MAPE). Here we list some details of our implemented baseline methods. For the hyper-parameters that are not mentioned, we adopt the default hyper-parameters from Shao et al. (2023):

**AGCRN** (Bai et al., 2020) We use two layers of AGCRN to capture the node-specific spatial and temporal dynamics. For the hyper-parameters, we set the hidden unit to 64 for all the AGCRN cells and the batch size also to 64. We set the learning rate to 0.003 and the embedding dimension to 64 for all 31 city datasets extracted from OSM+ dataset. Besides, we choose $L_1$ Loss as the loss function.

**Crossformer** (Zhang & Yan, 2022) When implementing Crossformer, we set segment length $L_{seg}$ to 24, as it is related to both the model performance and computation efficiency. Besides, we set the window size to 2. We use Adam optimizer with a 0.0002 learning rate and 0.0005 weight decay rate.

**DCRNN** (Li et al., 2017) Both encoder and decoder contain two recurrent layers. In each recurrent layer, there are 64 units and the initial learning rate is set to 0.003. Besides, the maximum step of random walks, i.e., $K$, is set to 2. For scheduled sampling, the thresholded inverse sigmoid function is used as the probability decay:

$$\epsilon_i = \frac{\tau}{\tau + \exp(i/\tau)} \tag{1}$$

where $i$ is the number of iterations while $\tau$ is the parameter to control the speed of convergence, which is set to 2,000 in the experiments.

**DLinear** (Diagne et al., 2012) For implementation details about DLinear, we adopt the default hyper-parameters from Shao et al. (2023) to train the models. The training epoch is set to 100.

**FEDformer** (Zhou et al., 2022) The FEDformer is trained using Adam optimizer with a learning rate of 0.0005. The batch size is set to 64. An early stopping counter is employed to stop the training process after three epochs if no loss degradation on the valid set is observed.

**GWNet** (DHANKHAR et al.) We use two layers of Graph WaveNet with a sequence of dilation factors {1, 2}. We randomly initialize node embeddings by a uniform distribution with a size of 10. We train our model using Adam optimizer with an initial learning rate of 0.0005. Dropout with $p = 0.3$ is applied to the outputs of the graph convolution layer.

## C.4 TRAFFIC POLICY EXPERIMENT

We use the script provided by CBData (Liang et al., 2023) to convert the roadnet file to the corresponding format required by CBEngine (Liang et al., 2023). Then, we conduct traffic flow simulation

experiments of 100,000 vehicles in six cities (Beijing, London, Los Angeles, New York, Paris and Shanghai). The simulation runs for a total of 3600 steps, with traffic flow data being introduced into the road at a uniform rate during the first 300 steps.

# D    EXTENDED EXPERIMENT RESULTS

To further illustrate supporting performance of the OSM+ dataset over traffic flow prediction task, we repeat each set of experiments five times and report their means and standard deviations on horizon 3, 6 and 12, following the common setting in this problem (Tedjopurnomo et al., 2020). Comprehensive results are shown in Table 7, Table 8 and Table 9.

Table 7: The performance comparison for seven baseline methods over 31 real-world city datasets with horizon=3. The best results in each row are in bold. All experiments are repeated five times, and the mean and standard deviation are reported.

| City | Metric | AGCRN | Crossformer | DCRNN | DLinear | FEDformer | GWNet | MTGNN |
|------|--------|-------|-------------|-------|---------|-----------|-------|-------|
| AA | MAE | $42.53 \pm 0.22$ | $\mathbf{39.89 \pm 0.11}$ | OOM | $40.56 \pm 0.00$ | $46.15 \pm 0.03$ | $40.65 \pm 0.19$ | $40.76 \pm 0.38$ |
| | RMSE | $87.82 \pm 0.22$ | $85.21 \pm 0.92$ | OOM | $82.00 \pm 0.00$ | $99.68 \pm 0.34$ | $\mathbf{81.14 \pm 1.17}$ | $83.04 \pm 0.33$ |
| | MAPE(%) | $38.97 \pm 0.18$ | $\mathbf{31.21 \pm 1.72}$ | OOM | $32.43 \pm 0.02$ | $39.76 \pm 0.10$ | $32.09 \pm 0.95$ | $33.27 \pm 2.39$ |
| BSL | MAE | $55.00 \pm 0.15$ | $49.76 \pm 0.96$ | $116.48 \pm 1.50$ | $\mathbf{49.12 \pm 0.00}$ | $52.55 \pm 0.18$ | $68.47 \pm 0.15$ | $65.86 \pm 0.60$ |
| | RMSE | $82.77 \pm 0.08$ | $\mathbf{74.24 \pm 2.83}$ | $155.13 \pm 0.43$ | $79.39 \pm 0.03$ | $76.74 \pm 0.37$ | $93.83 \pm 1.64$ | $90.52 \pm 1.20$ |
| | MAPE(%) | $45.85 \pm 0.35$ | $\mathbf{43.38 \pm 2.60}$ | $171.29 \pm 7.33$ | $42.93 \pm 0.01$ | $52.67 \pm 1.15$ | $91.08 \pm 3.34$ | $75.48 \pm 2.24$ |
| BRN | MAE | $46.82 \pm 1.30$ | $\mathbf{45.65 \pm 0.17}$ | OOM | $48.76 \pm 0.02$ | $51.16 \pm 0.10$ | $47.49 \pm 0.56$ | $59.36 \pm 0.86$ |
| | RMSE | $468.78 \pm 9.95$ | $\mathbf{454.56 \pm 1.64}$ | OOM | $454.98 \pm 0.18$ | $431.93 \pm 0.45$ | $440.49 \pm 7.76$ | $477.32 \pm 13.68$ |
| | MAPE(%) | $\mathbf{222.50 \pm 13.23}$ | $233.32 \pm 18.35$ | OOM | $229.36 \pm 2.05$ | $232.49 \pm 17.64$ | $278.14 \pm 19.92$ | $347.74 \pm 20.37$ |
| BHX | MAE | $88.84 \pm 0.35$ | $84.86 \pm 7.57$ | $289.70 \pm 11.26$ | $92.86 \pm 0.80$ | $104.66 \pm 0.44$ | $90.95 \pm 0.18$ | $\mathbf{84.13 \pm 0.64}$ |
| | RMSE | $137.05 \pm 0.61$ | $134.05 \pm 8.50$ | $425.16 \pm 37.43$ | $144.99 \pm 0.88$ | $158.84 \pm 0.19$ | $144.59 \pm 1.12$ | $\mathbf{130.56 \pm 0.88}$ |
| | MAPE(%) | $43.38 \pm 0.35$ | $\mathbf{40.51 \pm 2.58}$ | $191.60 \pm 57.29$ | $48.95 \pm 0.11$ | $52.14 \pm 0.90$ | $51.86 \pm 1.83$ | $45.67 \pm 3.76$ |
| BOL | MAE | $\mathbf{29.63 \pm 0.27}$ | $32.03 \pm 1.73$ | $32.74 \pm 0.38$ | $33.31 \pm 0.02$ | $34.19 \pm 0.04$ | $32.69 \pm 1.92$ | $29.68 \pm 0.23$ |
| | RMSE | $84.46 \pm 0.07$ | $84.23 \pm 0.25$ | $87.89 \pm 0.04$ | $83.81 \pm 0.06$ | $87.51 \pm 0.13$ | $84.37 \pm 0.56$ | $\mathbf{81.28 \pm 0.13}$ |
| | MAPE(%) | $\mathbf{18.44 \pm 1.09}$ | $19.69 \pm 0.44$ | $22.48 \pm 0.69$ | $30.71 \pm 0.10$ | $26.86 \pm 1.25$ | $22.51 \pm 0.53$ | $18.63 \pm 0.50$ |
| BOD | MAE | $62.34 \pm 0.22$ | $60.21 \pm 0.60$ | $244.58 \pm 20.30$ | $\mathbf{58.23 \pm 0.00}$ | $58.51 \pm 0.30$ | $59.78 \pm 0.05$ | $71.17 \pm 1.11$ |
| | RMSE | $100.52 \pm 0.47$ | $95.41 \pm 0.16$ | $324.45 \pm 24.77$ | $92.86 \pm 0.01$ | $\mathbf{92.09 \pm 0.27}$ | $93.00 \pm 0.09$ | $109.95 \pm 2.32$ |
| | MAPE(%) | $37.45 \pm 1.21$ | $\mathbf{33.47 \pm 0.43}$ | $280.19 \pm 14.03$ | $37.59 \pm 0.05$ | $38.28 \pm 0.84$ | $43.06 \pm 0.33$ | $45.75 \pm 0.42$ |
| BRE | MAE | $55.40 \pm 0.02$ | $56.33 \pm 1.40$ | OOM | $58.99 \pm 0.03$ | $58.80 \pm 0.04$ | $55.45 \pm 0.09$ | $\mathbf{55.18 \pm 0.17}$ |
| | RMSE | $92.18 \pm 0.08$ | $93.06 \pm 1.74$ | OOM | $97.06 \pm 0.11$ | $95.33 \pm 0.02$ | $\mathbf{91.31 \pm 0.11}$ | $91.72 \pm 0.48$ |
| | MAPE(%) | $36.08 \pm 0.26$ | $\mathbf{33.69 \pm 1.66}$ | OOM | $40.12 \pm 0.12$ | $40.41 \pm 0.49$ | $36.10 \pm 0.27$ | $34.93 \pm 0.22$ |
| KN | MAE | OOM | $39.17 \pm 1.36$ | $119.00 \pm 0.09$ | $\mathbf{35.47 \pm 0.00}$ | $38.35 \pm 0.37$ | $37.55 \pm 0.55$ | $40.06 \pm 0.38$ |
| | RMSE | OOM | $62.75 \pm 2.15$ | $152.84 \pm 0.67$ | $\mathbf{54.99 \pm 0.00}$ | $58.78 \pm 0.32$ | $58.21 \pm 0.67$ | $61.15 \pm 0.86$ |
| | MAPE(%) | OOM | $47.64 \pm 0.56$ | $293.93 \pm 0.88$ | $54.02 \pm 0.17$ | $61.77 \pm 1.36$ | $\mathbf{47.54 \pm 0.59}$ | $58.71 \pm 1.08$ |
| DA | MAE | $56.32 \pm 0.04$ | $51.12 \pm 0.79$ | OOM | $\mathbf{50.77 \pm 0.00}$ | $54.26 \pm 0.08$ | $51.80 \pm 0.11$ | $54.58 \pm 0.23$ |
| | RMSE | $88.65 \pm 0.20$ | $77.01 \pm 1.12$ | OOM | $\mathbf{75.41 \pm 0.00}$ | $79.78 \pm 0.25$ | $78.39 \pm 0.16$ | $86.54 \pm 1.08$ |
| | MAPE(%) | $50.79 \pm 0.00$ | $\mathbf{45.51 \pm 0.49}$ | OOM | $51.38 \pm 0.02$ | $58.16 \pm 0.89$ | $48.27 \pm 0.64$ | $48.10 \pm 0.11$ |
| ESS | MAE | $40.66 \pm 0.28$ | $36.95 \pm 0.10$ | $172.66 \pm 0.19$ | $41.06 \pm 0.00$ | $40.51 \pm 0.07$ | $36.86 \pm 0.08$ | $\mathbf{35.94 \pm 0.07}$ |
| | RMSE | $59.07 \pm 0.76$ | $53.29 \pm 0.27$ | $224.47 \pm 3.72$ | $59.39 \pm 0.01$ | $58.41 \pm 0.03$ | $53.72 \pm 0.22$ | $\mathbf{52.57 \pm 0.03}$ |
| | MAPE(%) | $35.54 \pm 1.10$ | $34.79 \pm 1.28$ | $302.81 \pm 15.18$ | $34.62 \pm 0.40$ | $36.01 \pm 0.71$ | $32.85 \pm 0.28$ | $\mathbf{31.51 \pm 2.29}$ |
| FRA | MAE | $124.69 \pm 0.05$ | $140.18 \pm 14.57$ | $185.87 \pm 2.29$ | $\mathbf{71.58 \pm 0.30}$ | $84.83 \pm 1.13$ | $139.06 \pm 1.84$ | $170.44 \pm 12.29$ |
| | RMSE | $157.09 \pm 0.09$ | $166.53 \pm 16.07$ | $237.20 \pm 2.82$ | $\mathbf{90.90 \pm 1.09}$ | $108.23 \pm 2.00$ | $163.25 \pm 2.09$ | $190.32 \pm 12.38$ |
| | MAPE(%) | $39.40 \pm 0.02$ | $41.18 \pm 3.19$ | $47.65 \pm 0.59$ | $\mathbf{20.21 \pm 0.07}$ | $23.41 \pm 0.15$ | $42.48 \pm 0.68$ | $51.70 \pm 3.62$ |
| GRZ | MAE | $59.06 \pm 0.02$ | $\mathbf{50.55 \pm 1.02}$ | $174.96 \pm 0.00$ | $52.27 \pm 0.00$ | $50.95 \pm 0.24$ | $55.07 \pm 0.74$ | $53.48 \pm 0.62$ |
| | RMSE | $89.07 \pm 0.02$ | $74.28 \pm 0.84$ | $220.49 \pm 0.00$ | $75.95 \pm 0.01$ | $\mathbf{74.23 \pm 0.17}$ | $81.10 \pm 1.00$ | $82.35 \pm 1.49$ |
| | MAPE(%) | $116.57 \pm 0.80$ | $68.30 \pm 4.29$ | $467.17 \pm 0.00$ | $\mathbf{63.68 \pm 0.15}$ | $66.96 \pm 2.12$ | $68.11 \pm 1.24$ | $71.41 \pm 0.40$ |
| GRQ | MAE | $63.24 \pm 0.01$ | $62.25 \pm 1.53$ | $162.87 \pm 0.00$ | $\mathbf{60.08 \pm 0.41}$ | $72.11 \pm 1.47$ | $64.33 \pm 0.16$ | $63.21 \pm 1.50$ |
| | RMSE | $87.02 \pm 0.08$ | $86.03 \pm 0.07$ | $218.69 \pm 0.00$ | $\mathbf{82.72 \pm 0.56}$ | $99.96 \pm 1.73$ | $88.41 \pm 0.03$ | $86.27 \pm 1.51$ |

| | | | | | | | |
|---|---|---|---|---|---|---|---|
| | MAPE(%) | 30.46 ± 0.19 | **28.82 ± 0.24** | 112.24 ± 0.00 | 31.41 ± 0.29 | 36.83 ± 0.57 | 31.79 ± 0.19 | 30.87 ± 1.08 |
| HAM | MAE | 45.68 ± 0.09 | 43.50 ± 0.10 | 97.51 ± 0.56 | 44.82 ± 0.00 | 46.34 ± 0.05 | **43.28 ± 0.01** | 44.05 ± 0.09 |
| | RMSE | 75.36 ± 0.60 | 70.76 ± 0.19 | 150.67 ± 2.84 | 73.13 ± 0.01 | 74.96 ± 0.18 | **70.69 ± 0.07** | 73.02 ± 0.19 |
| | MAPE(%) | 44.67 ± 0.36 | **42.79 ± 0.51** | 111.42 ± 5.09 | 47.93 ± 0.10 | 48.93 ± 0.17 | 43.48 ± 0.16 | 43.30 ± 0.13 |
| INN | MAE | 70.05 ± 0.15 | 67.34 ± 1.24 | 333.14 ± 2.31 | 76.43 ± 0.01 | 70.66 ± 0.65 | **65.65 ± 0.12** | OOM |
| | RMSE | 101.84 ± 0.05 | 97.03 ± 0.57 | 443.91 ± 2.79 | 113.93 ± 0.09 | 102.62 ± 0.80 | **95.28 ± 0.17** | OOM |
| | MAPE(%) | 31.38 ± 0.83 | 35.24 ± 5.44 | 304.36 ± 0.76 | 33.84 ± 0.45 | 30.70 ± 0.10 | **28.00 ± 0.42** | OOM |
| KS | MAE | 69.88 ± 0.37 | 73.22 ± 1.27 | 244.28 ± 0.00 | **63.22 ± 2.52** | 80.77 ± 0.65 | 63.34 ± 1.06 | 154.46 ± 1.90 |
| | RMSE | 203.93 ± 0.33 | 204.17 ± 2.19 | 342.80 ± 0.00 | 158.72 ± 4.21 | 177.02 ± 0.31 | **155.08 ± 0.53** | 233.89 ± 3.73 |
| | MAPE(%) | **80.45 ± 1.04** | 87.42 ± 3.53 | 440.12 ± 0.00 | 81.40 ± 4.85 | 105.62 ± 1.65 | 81.47 ± 2.36 | 235.91 ± 0.42 |
| MAN | MAE | 97.96 ± 0.71 | 87.80 ± 0.65 | 336.72 ± 0.00 | 92.36 ± 5.65 | 99.05 ± 1.29 | **84.26 ± 0.20** | 85.24 ± 0.93 |
| | RMSE | 169.67 ± 0.73 | 160.48 ± 1.62 | 448.89 ± 0.00 | 156.98 ± 7.85 | 167.30 ± 2.46 | **151.48 ± 0.49** | 154.15 ± 2.57 |
| | MAPE(%) | 39.66 ± 1.35 | 37.73 ± 0.54 | 283.06 ± 0.00 | 42.24 ± 2.20 | 46.21 ± 1.60 | **32.83 ± 0.26** | 36.21 ± 1.39 |
| MEL | MAE | 37.39 ± 0.00 | 36.21 ± 0.95 | OOM | 42.57 ± 0.85 | 40.80 ± 0.58 | 36.26 ± 0.01 | **35.86 ± 0.95** |
| | RMSE | 56.63 ± 0.01 | **54.30 ± 2.21** | OOM | 64.49 ± 1.03 | 60.80 ± 1.36 | 54.35 ± 0.06 | 54.78 ± 1.68 |
| | MAPE(%) | 37.60 ± 0.03 | 35.63 ± 0.72 | OOM | 37.82 ± 0.99 | 43.56 ± 0.04 | **27.22 ± 0.07** | 31.77 ± 0.80 |
| RTM | MAE | **49.08 ± 0.16** | 50.46 ± 0.06 | 170.03 ± 0.00 | 55.93 ± 0.01 | 58.45 ± 0.73 | 54.45 ± 0.28 | 51.45 ± 0.08 |
| | RMSE | **87.20 ± 0.12** | 87.80 ± 0.13 | 232.29 ± 0.00 | 95.05 ± 0.02 | 96.26 ± 0.90 | 92.68 ± 0.52 | 88.92 ± 0.36 |
| | MAPE(%) | **36.76 ± 0.05** | 41.26 ± 0.01 | 320.02 ± 0.00 | 44.61 ± 0.19 | 57.44 ± 1.81 | 43.72 ± 1.04 | 38.46 ± 1.26 |
| SDR | MAE | 88.07 ± 0.25 | 86.61 ± 1.76 | 259.05 ± 0.00 | 80.68 ± 0.02 | 107.62 ± 1.33 | **77.84 ± 0.41** | 78.50 ± 2.26 |
| | RMSE | 230.51 ± 0.94 | 231.19 ± 0.49 | 434.97 ± 0.00 | **187.89 ± 0.01** | 216.83 ± 2.05 | 202.47 ± 1.86 | 211.89 ± 0.85 |
| | MAPE(%) | 54.00 ± 1.15 | 53.10 ± 6.85 | 257.82 ± 0.00 | 43.89 ± 0.16 | 81.61 ± 0.53 | 44.22 ± 3.89 | **36.34 ± 0.17** |
| SP | MAE | 48.92 ± 0.05 | **47.51 ± 0.03** | 121.12 ± 2.04 | 50.74 ± 0.01 | 52.37 ± 0.11 | 47.84 ± 0.05 | 47.75 ± 0.11 |
| | RMSE | 70.45 ± 0.06 | **68.34 ± 0.07** | 172.55 ± 9.56 | 72.60 ± 0.06 | 74.68 ± 0.06 | 68.91 ± 0.17 | 68.90 ± 0.02 |
| | MAPE(%) | 39.84 ± 0.53 | **36.77 ± 0.11** | 102.07 ± 25.06 | 43.49 ± 0.00 | 43.79 ± 0.59 | 37.92 ± 0.33 | 37.04 ± 0.13 |
| SXB | MAE | 76.69 ± 0.07 | 73.93 ± 0.14 | 259.67 ± 0.00 | 78.84 ± 0.05 | 80.54 ± 0.24 | 74.23 ± 0.03 | **73.87 ± 0.41** |
| | RMSE | 134.85 ± 0.14 | **130.71 ± 0.32** | 360.44 ± 0.00 | 138.48 ± 0.03 | 141.88 ± 0.15 | 131.69 ± 0.05 | 131.14 ± 0.23 |
| | MAPE(%) | 40.04 ± 1.09 | 40.73 ± 2.09 | 223.03 ± 0.00 | 43.50 ± 0.18 | 43.51 ± 0.74 | 37.64 ± 0.26 | **36.53 ± 0.63** |
| STR | MAE | 57.40 ± 0.43 | 57.84 ± 3.60 | 59.86 ± 1.67 | 60.07 ± 0.08 | 68.33 ± 1.18 | **55.70 ± 0.02** | OOM |
| | RMSE | 74.79 ± 0.28 | 75.55 ± 4.24 | 78.25 ± 3.02 | 78.53 ± 0.16 | 89.06 ± 1.27 | **72.13 ± 0.07** | OOM |
| | MAPE(%) | 18.44 ± 0.08 | 19.11 ± 2.18 | 19.07 ± 0.70 | 20.01 ± 0.01 | 22.32 ± 0.45 | **18.10 ± 0.06** | OOM |
| TPE | MAE | 126.81 ± 0.07 | 126.02 ± 3.81 | 490.25 ± 8.98 | 125.41 ± 0.01 | 134.61 ± 0.09 | **117.30 ± 0.40** | 121.00 ± 0.90 |
| | RMSE | 555.59 ± 1.23 | 561.67 ± 10.18 | 988.24 ± 11.03 | **482.59 ± 0.38** | 540.48 ± 5.15 | 493.02 ± 0.48 | 512.90 ± 1.36 |
| | MAPE(%) | 42.20 ± 0.35 | 45.74 ± 5.09 | 266.56 ± 4.39 | 41.80 ± 0.12 | 47.72 ± 0.55 | **38.30 ± 1.75** | 40.01 ± 1.54 |
| TO | MAE | 77.75 ± 0.14 | 71.06 ± 0.96 | 313.88 ± 0.21 | **68.52 ± 0.01** | 74.48 ± 0.07 | 74.91 ± 1.05 | 80.07 ± 1.05 |
| | RMSE | 131.86 ± 0.13 | 114.97 ± 2.10 | 415.51 ± 1.03 | **111.00 ± 0.07** | 118.37 ± 0.35 | 116.63 ± 0.82 | 124.72 ± 1.31 |
| | MAPE(%) | 50.86 ± 0.32 | **39.42 ± 0.29** | 400.19 ± 7.19 | 40.27 ± 0.47 | 48.37 ± 0.98 | 46.33 ± 2.97 | 52.21 ± 2.04 |
| YTO | MAE | 45.81 ± 0.12 | 45.25 ± 0.40 | 129.55 ± 18.89 | 59.49 ± 0.01 | 52.16 ± 0.04 | 46.68 ± 0.25 | **44.65 ± 0.07** |
| | RMSE | 77.29 ± 0.30 | **74.48 ± 0.06** | 192.41 ± 21.36 | 95.84 ± 0.09 | 81.85 ± 0.04 | 77.47 ± 0.47 | 75.03 ± 0.00 |
| | MAPE(%) | 33.49 ± 0.02 | **32.72 ± 1.55** | 114.20 ± 27.77 | 42.70 ± 0.66 | 46.45 ± 1.69 | 35.18 ± 0.38 | 33.36 ± 2.07 |
| TLS | MAE | 257.32 ± 0.36 | **255.21 ± 0.28** | 264.24 ± 6.15 | 263.84 ± 0.00 | 296.07 ± 0.02 | 255.32 ± 0.00 | 259.00 ± 0.03 |
| | RMSE | 349.78 ± 0.86 | 342.51 ± 0.43 | 352.70 ± 9.62 | 348.15 ± 0.03 | 410.70 ± 0.70 | **341.22 ± 0.34** | 349.09 ± 0.86 |
| | MAPE(%) | 754.88 ± 3.75 | 746.38 ± 4.66 | 791.79 ± 77.77 | 872.83 ± 0.91 | 836.22 ± 1.06 | 747.55 ± 5.53 | **726.09 ± 9.80** |
| UTC | MAE | OOM | 51.67 ± 0.80 | OOM | 44.40 ± 1.21 | 61.74 ± 1.37 | 75.85 ± 0.25 | **39.61 ± 0.20** |
| | RMSE | OOM | 83.26 ± 3.39 | OOM | 75.24 ± 1.49 | 91.06 ± 0.40 | 120.97 ± 0.66 | **68.41 ± 0.37** |
| | MAPE(%) | OOM | 57.21 ± 15.30 | OOM | 45.37 ± 1.44 | 77.81 ± 4.77 | 91.81 ± 5.44 | **37.55 ± 2.65** |
| VNO | MAE | 82.75 ± 0.08 | 77.98 ± 0.27 | OOM | **69.08 ± 1.53** | 81.57 ± 0.28 | 69.25 ± 0.23 | 89.80 ± 1.00 |
| | RMSE | 112.24 ± 0.29 | 106.39 ± 0.56 | OOM | **94.18 ± 2.13** | 110.65 ± 0.82 | 94.42 ± 0.06 | 118.34 ± 1.00 |
| | MAPE(%) | 47.89 ± 0.26 | 42.38 ± 0.14 | OOM | 35.59 ± 0.75 | 42.38 ± 0.60 | **35.32 ± 0.81** | 57.14 ± 1.75 |
| WOB | MAE | 52.66 ± 0.05 | 51.73 ± 1.69 | 54.51 ± 0.05 | 56.05 ± 0.02 | 54.89 ± 0.01 | 51.27 ± 0.18 | **51.12 ± 0.01** |
| | RMSE | 81.81 ± 0.31 | 79.60 ± 3.58 | 84.03 ± 0.30 | 85.65 ± 0.02 | 82.31 ± 0.02 | **78.15 ± 0.24** | 78.27 ± 0.04 |
| | MAPE(%) | 40.76 ± 0.38 | **37.38 ± 0.17** | 42.23 ± 0.24 | 46.71 ± 0.06 | 46.64 ± 0.10 | 39.56 ± 0.58 | 39.41 ± 0.80 |
| ZRH | MAE | OOM | 53.23 ± 0.05 | OOM | 56.33 ± 0.00 | 56.72 ± 0.08 | 59.24 ± 7.75 | **52.34 ± 0.18** |

| | RMSE | OOM | 75.13 ± 0.05 | OOM | 79.32 ± 0.01 | 79.36 ± 0.01 | 83.96 ± 11.26 | **74.05 ± 0.34** |
| | MAPE(%) | OOM | 35.77 ± 1.18 | OOM | 40.87 ± 0.04 | 41.31 ± 0.53 | 44.88 ± 11.14 | **34.95 ± 0.46** |

Table 8: The performance comparison for seven baseline methods over 31 real-world city datasets with horizon=6. The best results in each row are in bold. All experiments are repeated five times, and the mean and standard deviation are reported.

| City | Metric | AGCRN | Crossformer | DCRNN | DLinear | FEDformer | GWNet | MTGNN |
|------|--------|-------|-------------|-------|---------|-----------|-------|-------|
| AA | MAE | 47.27 ± 0.41 | **43.29 ± 0.11** | OOM | 46.25 ± 0.00 | 50.21 ± 0.02 | 46.22 ± 0.59 | 45.61 ± 0.55 |
| | RMSE | 100.54 ± 0.22 | **95.50 ± 1.21** | OOM | 97.90 ± 0.03 | 112.57 ± 0.37 | 97.17 ± 2.27 | 97.30 ± 0.43 |
| | MAPE(%) | 43.35 ± 0.73 | **33.68 ± 0.45** | OOM | 36.33 ± 0.15 | 42.69 ± 0.09 | 36.93 ± 1.56 | 37.46 ± 3.09 |
| BSL | MAE | 63.95 ± 0.02 | 61.81 ± 3.57 | 118.10 ± 0.84 | **59.01 ± 0.01** | 59.94 ± 1.42 | 81.55 ± 1.76 | 78.88 ± 1.13 |
| | RMSE | 94.86 ± 0.09 | 91.78 ± 4.55 | 157.37 ± 1.12 | 99.52 ± 0.04 | **87.06 ± 2.86** | 114.99 ± 4.41 | 110.81 ± 2.49 |
| | MAPE(%) | 54.81 ± 0.20 | 68.11 ± 2.43 | 175.12 ± 9.70 | **49.76 ± 0.16** | 59.65 ± 0.08 | 109.78 ± 3.98 | 91.64 ± 2.08 |
| BRN | MAE | 49.20 ± 1.52 | **48.47 ± 0.32** | OOM | 51.65 ± 0.01 | 54.01 ± 0.18 | 50.27 ± 0.10 | 69.27 ± 0.82 |
| | RMSE | 481.50 ± 16.19 | 465.00 ± 1.10 | OOM | 464.66 ± 0.33 | **450.70 ± 0.37** | 468.88 ± 0.39 | 508.09 ± 12.22 |
| | MAPE(%) | 216.94 ± 6.34 | **186.02 ± 20.85** | OOM | 244.39 ± 5.93 | 242.43 ± 19.54 | 293.55 ± 20.22 | 411.51 ± 30.03 |
| BHX | MAE | 101.87 ± 0.00 | **86.82 ± 7.11** | 299.18 ± 30.69 | 103.14 ± 2.28 | 113.03 ± 0.16 | 99.17 ± 0.39 | 89.68 ± 0.34 |
| | RMSE | 148.33 ± 0.23 | **133.22 ± 9.67** | 426.46 ± 61.57 | 156.29 ± 2.41 | 175.26 ± 0.51 | 150.67 ± 0.76 | 137.11 ± 1.22 |
| | MAPE(%) | 59.01 ± 0.56 | **45.71 ± 1.30** | 229.60 ± 44.64 | 57.28 ± 1.81 | 57.12 ± 0.60 | 56.87 ± 5.04 | 49.85 ± 1.96 |
| BOL | MAE | **30.99 ± 0.09** | 32.53 ± 1.31 | 37.09 ± 0.61 | 35.99 ± 0.07 | 35.71 ± 0.38 | 33.47 ± 0.60 | 31.57 ± 0.32 |
| | RMSE | 87.74 ± 0.14 | 87.00 ± 0.68 | 93.43 ± 0.01 | 88.23 ± 0.03 | 87.93 ± 0.81 | 82.40 ± 0.12 | **82.26 ± 0.01** |
| | MAPE(%) | **19.58 ± 1.26** | 20.30 ± 0.26 | 26.38 ± 0.99 | 33.56 ± 0.24 | 29.66 ± 1.49 | 28.79 ± 0.72 | 21.63 ± 0.37 |
| BOD | MAE | 69.25 ± 0.17 | 65.32 ± 0.92 | 242.05 ± 12.05 | **64.43 ± 0.01** | 64.68 ± 0.20 | 70.55 ± 0.22 | 86.45 ± 0.91 |
| | RMSE | 112.27 ± 0.50 | 103.95 ± 0.50 | 321.50 ± 16.12 | 102.17 ± 0.01 | **101.21 ± 0.16** | 108.43 ± 0.09 | 135.49 ± 2.53 |
| | MAPE(%) | 39.52 ± 1.05 | **35.61 ± 0.49** | 280.80 ± 1.26 | 42.14 ± 0.23 | 42.19 ± 0.72 | 52.86 ± 0.88 | 55.08 ± 1.37 |
| BRE | MAE | 56.30 ± 0.01 | 56.94 ± 0.75 | OOM | 62.32 ± 0.01 | 60.38 ± 0.08 | 56.86 ± 0.01 | **56.26 ± 0.27** |
| | RMSE | 93.86 ± 0.02 | 94.36 ± 0.99 | OOM | 101.83 ± 0.07 | 97.83 ± 0.11 | 93.62 ± 0.02 | **93.58 ± 0.67** |
| | MAPE(%) | 36.30 ± 0.14 | **34.78 ± 0.07** | OOM | 41.95 ± 0.28 | 41.36 ± 0.48 | 37.02 ± 0.02 | 35.45 ± 0.29 |
| KN | MAE | OOM | 43.36 ± 1.67 | 118.57 ± 2.65 | **37.85 ± 0.00** | 40.09 ± 0.24 | 43.01 ± 1.18 | 45.06 ± 0.30 |
| | RMSE | OOM | 69.19 ± 2.16 | 151.98 ± 2.99 | 61.54 ± 0.01 | **61.47 ± 0.14** | 68.27 ± 1.68 | 68.58 ± 0.14 |
| | MAPE(%) | OOM | **47.64 ± 0.45** | 294.00 ± 8.92 | 59.41 ± 0.17 | 64.05 ± 1.65 | 53.57 ± 1.58 | 68.56 ± 3.07 |
| DA | MAE | 58.34 ± 0.14 | **53.03 ± 0.05** | OOM | 53.93 ± 0.01 | 56.35 ± 0.07 | 55.08 ± 0.19 | 59.16 ± 0.15 |
| | RMSE | 92.05 ± 0.44 | **80.86 ± 0.21** | OOM | 81.10 ± 0.07 | 83.79 ± 0.23 | 84.75 ± 0.59 | 96.79 ± 0.57 |
| | MAPE(%) | 51.28 ± 0.03 | **47.58 ± 0.26** | OOM | 53.46 ± 0.08 | 59.83 ± 0.81 | 50.41 ± 1.14 | 50.52 ± 0.23 |
| ESS | MAE | 41.58 ± 0.11 | 39.69 ± 0.44 | 173.69 ± 0.75 | 47.27 ± 0.00 | 43.48 ± 0.01 | 38.23 ± 0.14 | **37.59 ± 0.06** |
| | RMSE | 61.02 ± 0.33 | 57.73 ± 0.93 | 226.42 ± 2.28 | 68.86 ± 0.02 | 63.05 ± 0.26 | 56.05 ± 0.23 | **55.76 ± 0.09** |
| | MAPE(%) | 34.29 ± 0.34 | 40.50 ± 4.51 | 299.82 ± 15.01 | 40.99 ± 0.15 | 39.62 ± 0.40 | 34.16 ± 0.17 | **34.01 ± 2.44** |
| FRA | MAE | 158.46 ± 0.09 | 141.89 ± 33.64 | 187.73 ± 7.77 | **93.49 ± 1.42** | 107.62 ± 2.29 | 173.62 ± 3.08 | 258.38 ± 13.05 |
| | RMSE | 192.87 ± 0.02 | 172.02 ± 35.30 | 236.87 ± 10.11 | **115.65 ± 1.17** | 136.75 ± 3.69 | 202.44 ± 2.63 | 279.20 ± 13.65 |
| | MAPE(%) | 52.05 ± 0.01 | 43.28 ± 8.91 | 52.30 ± 1.53 | **27.45 ± 0.40** | 30.92 ± 0.64 | 54.62 ± 1.16 | 81.61 ± 3.79 |
| GRZ | MAE | 60.79 ± 0.01 | **52.70 ± 0.96** | 185.12 ± 0.00 | 58.29 ± 0.01 | 54.03 ± 0.24 | 58.32 ± 0.87 | 56.03 ± 0.20 |
| | RMSE | 91.77 ± 0.08 | **77.53 ± 0.71** | 233.72 ± 0.00 | 84.30 ± 0.04 | 78.69 ± 0.12 | 86.47 ± 1.17 | 87.69 ± 0.61 |
| | MAPE(%) | 110.69 ± 1.77 | **65.44 ± 6.48** | 465.95 ± 0.00 | 72.50 ± 1.27 | 69.79 ± 3.62 | 71.75 ± 1.36 | 72.35 ± 0.12 |
| GRQ | MAE | 68.43 ± 0.01 | 65.61 ± 1.26 | 161.01 ± 0.00 | **64.78 ± 0.91** | 79.46 ± 0.32 | 66.49 ± 0.22 | 71.04 ± 3.46 |
| | RMSE | 93.25 ± 0.24 | 90.55 ± 0.25 | 217.62 ± 0.00 | **89.18 ± 1.54** | 110.72 ± 0.49 | 91.41 ± 0.15 | 95.64 ± 4.02 |
| | MAPE(%) | 33.78 ± 0.31 | **30.45 ± 1.73** | 114.12 ± 0.00 | 35.59 ± 0.64 | 41.81 ± 0.46 | 33.33 ± 0.42 | 36.21 ± 1.79 |
| HAM | MAE | 46.44 ± 0.12 | 44.26 ± 0.07 | 97.38 ± 0.48 | 46.26 ± 0.01 | 47.54 ± 0.10 | **44.16 ± 0.01** | 45.04 ± 0.01 |
| | RMSE | 77.82 ± 0.73 | 74.06 ± 0.10 | 150.66 ± 2.47 | 77.55 ± 0.01 | 79.25 ± 0.22 | **74.06 ± 0.10** | 79.08 ± 1.23 |
| | MAPE(%) | 45.57 ± 0.70 | 44.25 ± 2.04 | 111.18 ± 3.96 | 49.44 ± 0.06 | 50.07 ± 0.06 | **43.68 ± 0.20** | 44.06 ± 0.07 |
| INN | MAE | 71.83 ± 0.48 | 67.78 ± 0.30 | 337.43 ± 6.70 | 86.47 ± 0.02 | 73.73 ± 0.55 | **66.95 ± 0.28** | OOM |
| | RMSE | 104.72 ± 0.59 | 98.77 ± 1.04 | 452.25 ± 0.27 | 133.76 ± 0.11 | 107.19 ± 0.71 | **97.41 ± 0.45** | OOM |

| | | | | | | | | |
|---|---|---|---|---|---|---|---|---|
| | MAPE(%) | 29.79 ± 1.13 | 34.04 ± 3.08 | 294.18 ± 24.63 | 37.85 ± 0.91 | 32.08 ± 0.02 | **28.82 ± 1.09** | OOM |
| KS | MAE | 77.06 ± 0.53 | 83.40 ± 5.34 | 229.47 ± 0.00 | 72.68 ± 1.41 | 85.40 ± 0.05 | **70.79 ± 0.35** | 190.66 ± 2.64 |
| | RMSE | 216.34 ± 0.41 | 218.00 ± 4.66 | 332.48 ± 0.00 | 177.06 ± 2.66 | 187.31 ± 0.67 | **174.04 ± 0.28** | 277.37 ± 5.51 |
| | MAPE(%) | 97.13 ± 1.41 | 111.00 ± 17.20 | 410.19 ± 0.00 | 99.24 ± 1.87 | 116.48 ± 0.61 | **93.42 ± 0.13** | 308.03 ± 0.63 |
| MAN | MAE | 106.44 ± 1.13 | 94.31 ± 0.83 | 335.71 ± 0.00 | 103.64 ± 10.23 | 110.65 ± 0.92 | **93.95 ± 1.00** | 94.68 ± 2.13 |
| | RMSE | 180.81 ± 1.54 | 169.06 ± 0.26 | 448.79 ± 0.00 | 172.31 ± 13.89 | 184.62 ± 2.27 | **163.00 ± 3.21** | 168.90 ± 4.47 |
| | MAPE(%) | 43.13 ± 1.40 | 43.34 ± 2.45 | 279.02 ± 0.00 | 47.57 ± 4.43 | 51.47 ± 0.57 | **37.68 ± 0.64** | 42.02 ± 1.16 |
| MEL | MAE | 48.56 ± 0.00 | **43.98 ± 1.73** | OOM | 58.89 ± 0.02 | 50.66 ± 0.49 | 49.02 ± 0.20 | 46.12 ± 1.29 |
| | RMSE | 75.34 ± 0.01 | **66.14 ± 3.55** | OOM | 89.81 ± 0.46 | 76.23 ± 0.70 | 74.99 ± 0.31 | 72.62 ± 2.27 |
| | MAPE(%) | 44.66 ± 0.05 | 40.38 ± 0.06 | OOM | 54.22 ± 0.50 | 53.21 ± 0.56 | **33.97 ± 0.06** | 39.97 ± 1.00 |
| RTM | MAE | **51.36 ± 0.14** | 52.84 ± 1.05 | 179.34 ± 0.00 | 64.94 ± 0.05 | 65.13 ± 0.38 | 63.44 ± 0.77 | 55.79 ± 0.08 |
| | RMSE | 92.19 ± 0.07 | **91.97 ± 0.65** | 240.67 ± 0.00 | 110.95 ± 0.04 | 106.66 ± 0.68 | 106.73 ± 1.16 | 97.29 ± 0.25 |
| | MAPE(%) | **39.55 ± 0.86** | 48.88 ± 3.29 | 349.98 ± 0.00 | 50.56 ± 0.21 | 63.69 ± 0.70 | 49.48 ± 0.75 | 40.87 ± 1.70 |
| SDR | MAE | 101.26 ± 0.42 | 98.18 ± 2.76 | 256.61 ± 0.00 | 93.27 ± 0.06 | 119.78 ± 2.36 | **86.47 ± 0.70** | 91.34 ± 2.73 |
| | RMSE | 252.60 ± 0.01 | 249.70 ± 1.91 | 433.47 ± 0.00 | **214.37 ± 0.03** | 239.59 ± 2.95 | 220.15 ± 1.88 | 235.42 ± 0.34 |
| | MAPE(%) | 58.36 ± 1.09 | 52.17 ± 5.47 | 255.29 ± 0.00 | 51.78 ± 0.42 | 89.07 ± 2.29 | 49.86 ± 5.91 | **41.60 ± 0.36** |
| SP | MAE | 49.19 ± 0.05 | **47.78 ± 0.02** | 122.39 ± 4.19 | 52.40 ± 0.01 | 53.30 ± 0.09 | 48.28 ± 0.09 | 48.06 ± 0.17 |
| | RMSE | 70.83 ± 0.02 | **68.35 ± 0.03** | 175.09 ± 12.25 | 75.17 ± 0.04 | 75.91 ± 0.14 | 69.62 ± 0.23 | 69.28 ± 0.15 |
| | MAPE(%) | 39.89 ± 0.29 | 39.42 ± 0.10 | 102.80 ± 22.32 | 44.96 ± 0.02 | 44.63 ± 0.37 | 37.82 ± 0.32 | **37.33 ± 0.28** |
| SXB | MAE | 78.07 ± 0.07 | **75.86 ± 0.22** | 261.26 ± 0.00 | 83.83 ± 0.05 | 83.48 ± 0.26 | 76.43 ± 0.07 | 76.10 ± 0.20 |
| | RMSE | 137.75 ± 0.13 | **134.97 ± 1.08** | 361.96 ± 0.00 | 146.93 ± 0.03 | 147.22 ± 0.28 | 135.84 ± 0.06 | 136.00 ± 0.61 |
| | MAPE(%) | 39.38 ± 0.52 | 36.88 ± 0.64 | 223.02 ± 0.00 | 46.07 ± 0.29 | 44.95 ± 0.49 | 38.86 ± 0.28 | **36.87 ± 0.37** |
| STR | MAE | 58.31 ± 0.07 | 55.90 ± 0.33 | 65.43 ± 3.82 | 63.52 ± 0.53 | 67.70 ± 0.29 | **55.29 ± 0.12** | OOM |
| | RMSE | 75.53 ± 0.02 | 72.34 ± 0.67 | 86.65 ± 5.08 | 82.43 ± 0.47 | 86.71 ± 0.16 | **71.79 ± 0.33** | OOM |
| | MAPE(%) | 20.16 ± 0.01 | **18.61 ± 0.29** | 21.85 ± 1.88 | 22.95 ± 0.11 | 24.03 ± 0.61 | 18.78 ± 0.04 | OOM |
| TPE | MAE | 134.51 ± 0.20 | 130.49 ± 1.84 | 509.35 ± 13.02 | 138.44 ± 0.00 | 144.56 ± 0.25 | **126.48 ± 0.29** | 129.85 ± 0.99 |
| | RMSE | 604.37 ± 1.42 | 606.52 ± 6.88 | 1002.98 ± 12.83 | 592.89 ± 0.12 | 616.55 ± 3.87 | **567.78 ± 0.27** | 589.03 ± 1.28 |
| | MAPE(%) | 45.86 ± 0.49 | 43.77 ± 0.72 | 285.05 ± 4.83 | 44.76 ± 0.06 | 50.48 ± 0.57 | **40.21 ± 1.83** | 41.77 ± 1.29 |
| TO | MAE | 87.26 ± 0.28 | **77.97 ± 0.72** | 315.29 ± 0.01 | 80.61 ± 0.02 | 83.05 ± 0.09 | 95.23 ± 1.17 | 97.61 ± 0.70 |
| | RMSE | 149.60 ± 0.47 | **128.32 ± 1.66** | 421.69 ± 3.47 | 134.94 ± 0.10 | 133.55 ± 0.26 | 150.90 ± 1.60 | 154.19 ± 0.30 |
| | MAPE(%) | 55.62 ± 0.22 | **43.26 ± 0.12** | 390.80 ± 15.64 | 46.08 ± 0.69 | 53.08 ± 1.63 | 56.75 ± 2.22 | 63.98 ± 1.64 |
| YTO | MAE | 52.35 ± 0.07 | **51.04 ± 0.30** | 148.49 ± 17.87 | 86.46 ± 0.04 | 60.73 ± 0.19 | 57.26 ± 0.38 | 51.26 ± 0.14 |
| | RMSE | 88.15 ± 0.01 | **83.84 ± 0.62** | 219.71 ± 22.22 | 137.24 ± 0.27 | 95.65 ± 0.01 | 95.71 ± 0.74 | 86.95 ± 0.04 |
| | MAPE(%) | 38.58 ± 0.02 | 37.69 ± 3.84 | 110.54 ± 44.26 | 65.08 ± 1.92 | 53.94 ± 2.03 | 37.79 ± 0.83 | **37.62 ± 1.48** |
| TLS | MAE | 257.69 ± 0.41 | **255.12 ± 0.04** | 264.27 ± 6.31 | 263.95 ± 0.00 | 294.82 ± 0.19 | 255.35 ± 0.01 | 259.64 ± 0.41 |
| | RMSE | 350.19 ± 1.23 | 342.33 ± 0.54 | 352.67 ± 9.87 | 348.26 ± 0.09 | 408.19 ± 0.21 | **340.68 ± 0.33** | 351.37 ± 2.05 |
| | MAPE(%) | 761.22 ± 0.81 | 745.90 ± 7.41 | 792.14 ± 78.55 | 869.43 ± 1.90 | 833.62 ± 1.57 | 749.50 ± 6.72 | **730.89 ± 9.58** |
| UTC | MAE | OOM | 66.56 ± 26.30 | OOM | 48.87 ± 0.11 | 68.00 ± 0.85 | 74.93 ± 0.19 | **40.09 ± 0.10** |
| | RMSE | OOM | 95.10 ± 17.24 | OOM | 83.02 ± 0.25 | 99.18 ± 0.36 | 122.75 ± 1.30 | **72.37 ± 0.03** |
| | MAPE(%) | OOM | 104.15 ± 72.84 | OOM | 51.11 ± 0.40 | 90.66 ± 5.26 | 89.98 ± 4.19 | **37.86 ± 3.07** |
| VNO | MAE | 87.69 ± 0.14 | 83.21 ± 0.40 | OOM | 74.00 ± 1.45 | 86.63 ± 0.41 | **72.98 ± 0.37** | 96.27 ± 0.67 |
| | RMSE | 118.99 ± 0.33 | 113.23 ± 0.07 | OOM | 100.58 ± 1.74 | 118.15 ± 0.43 | **99.91 ± 0.11** | 127.62 ± 0.73 |
| | MAPE(%) | 53.52 ± 0.14 | 48.97 ± 1.46 | OOM | 41.17 ± 0.65 | 46.50 ± 0.59 | **38.67 ± 1.02** | 64.60 ± 1.76 |
| WOB | MAE | 53.64 ± 0.29 | 54.06 ± 3.31 | 58.06 ± 0.03 | 61.11 ± 0.00 | 56.56 ± 0.08 | 53.34 ± 0.36 | **52.71 ± 0.12** |
| | RMSE | 84.29 ± 0.58 | 83.78 ± 6.06 | 91.34 ± 0.32 | 95.94 ± 0.02 | 85.20 ± 0.03 | 82.52 ± 0.55 | **81.89 ± 0.16** |
| | MAPE(%) | 40.44 ± 0.19 | **39.46 ± 1.23** | 45.09 ± 0.29 | 49.95 ± 0.02 | 48.87 ± 0.60 | 41.26 ± 0.69 | 40.60 ± 1.35 |
| ZRH | MAE | OOM | 54.55 ± 0.39 | OOM | 59.12 ± 0.01 | 58.25 ± 0.02 | 62.40 ± 10.21 | **53.08 ± 0.23** |
| | RMSE | OOM | 77.09 ± 0.55 | OOM | 83.71 ± 0.00 | 81.89 ± 0.11 | 89.30 ± 15.55 | **75.22 ± 0.44** |
| | MAPE(%) | OOM | 35.78 ± 2.65 | OOM | 43.08 ± 0.12 | 42.26 ± 0.44 | 46.00 ± 11.94 | **35.18 ± 0.34** |

Table 9: The performance comparison for seven baseline methods over 31 real-world city datasets with horizon=12. The best results in each row are in bold. All experiments are repeated five times, and the mean and standard deviation are reported.

| City | Metric | AGCRN | Crossformer | DCRNN | DLinear | FEDformer | GWNet | MTGNN |
|------|--------|-------|-------------|-------|---------|-----------|-------|-------|
| AA | MAE | 55.05 ± 0.93 | **49.30 ± 0.79** | OOM | 56.58 ± 0.01 | 59.64 ± 0.20 | 55.81 ± 1.34 | 53.30 ± 0.80 |
| | RMSE | 118.56 ± 0.58 | **110.08 ± 0.06** | OOM | 122.58 ± 0.10 | 132.58 ± 0.19 | 117.78 ± 3.74 | 115.28 ± 0.52 |
| | MAPE(%) | 51.73 ± 1.84 | **41.13 ± 2.19** | OOM | 44.54 ± 0.12 | 53.19 ± 0.31 | 46.17 ± 2.80 | 44.31 ± 3.53 |
| BSL | MAE | 74.87 ± 0.54 | 81.11 ± 3.35 | 120.92 ± 0.42 | 76.52 ± 0.22 | **66.00 ± 0.70** | 97.61 ± 4.34 | 89.52 ± 0.36 |
| | RMSE | 110.37 ± 0.82 | 122.17 ± 4.15 | 160.46 ± 2.57 | 126.12 ± 0.48 | **96.52 ± 0.35** | 139.03 ± 8.66 | 126.94 ± 0.56 |
| | MAPE(%) | 65.54 ± 0.65 | 89.04 ± 5.12 | 183.70 ± 15.83 | **62.84 ± 0.30** | 69.59 ± 3.62 | 129.70 ± 6.69 | 111.43 ± 0.24 |
| BRN | MAE | **53.50 ± 2.10** | 54.20 ± 1.85 | OOM | 56.13 ± 0.01 | 61.14 ± 0.33 | 54.13 ± 0.47 | 84.36 ± 1.26 |
| | RMSE | 500.59 ± 21.34 | 483.08 ± 0.25 | OOM | 495.21 ± 0.17 | 489.57 ± 0.68 | **479.82 ± 1.60** | 553.53 ± 12.87 |
| | MAPE(%) | 245.52 ± 4.25 | **202.93 ± 62.89** | OOM | 290.64 ± 1.52 | 307.33 ± 16.91 | 340.44 ± 23.69 | 513.55 ± 31.06 |
| BHX | MAE | 146.03 ± 0.35 | 102.03 ± 15.50 | 282.12 ± 13.73 | 131.89 ± 9.65 | 140.17 ± 0.70 | 129.19 ± 2.55 | **101.51 ± 0.10** |
| | RMSE | 208.28 ± 0.65 | 155.04 ± 22.46 | 384.96 ± 40.10 | 193.75 ± 6.56 | 214.60 ± 0.16 | 192.02 ± 0.76 | **154.51 ± 0.86** |
| | MAPE(%) | 109.60 ± 0.28 | 63.90 ± 4.44 | 279.89 ± 61.60 | 90.05 ± 7.98 | 86.17 ± 0.27 | 80.07 ± 9.65 | **58.53 ± 2.38** |
| BOL | MAE | **33.38 ± 0.12** | 35.32 ± 0.68 | 45.53 ± 0.92 | 42.42 ± 0.09 | 44.03 ± 0.11 | 39.58 ± 2.22 | 34.94 ± 0.49 |
| | RMSE | 93.57 ± 0.67 | 93.44 ± 2.09 | 106.98 ± 0.01 | 102.71 ± 0.26 | 105.61 ± 0.31 | 97.56 ± 0.15 | **92.74 ± 0.59** |
| | MAPE(%) | 23.09 ± 0.63 | **22.13 ± 0.46** | 33.86 ± 1.51 | 38.07 ± 0.13 | 33.57 ± 0.61 | 27.38 ± 1.10 | 22.27 ± 0.29 |
| BOD | MAE | 84.17 ± 0.76 | **78.48 ± 2.18** | 234.94 ± 3.51 | 78.76 ± 0.01 | 87.29 ± 0.40 | 91.20 ± 1.25 | 109.35 ± 1.40 |
| | RMSE | 138.13 ± 1.65 | 125.87 ± 0.23 | 312.43 ± 6.74 | **122.38 ± 0.08** | 131.87 ± 0.90 | 139.59 ± 1.34 | 173.27 ± 0.92 |
| | MAPE(%) | 44.29 ± 0.82 | **41.19 ± 1.49** | 278.32 ± 1.47 | 54.23 ± 0.18 | 59.39 ± 0.18 | 73.00 ± 2.49 | 74.25 ± 5.28 |
| BRE | MAE | **57.33 ± 0.09** | 59.38 ± 0.08 | OOM | 68.48 ± 0.01 | 64.98 ± 0.09 | 58.62 ± 0.06 | 58.05 ± 0.39 |
| | RMSE | **95.67 ± 0.22** | 98.02 ± 0.06 | OOM | 111.31 ± 0.02 | 103.81 ± 0.10 | 96.37 ± 0.06 | 96.10 ± 0.84 |
| | MAPE(%) | 36.89 ± 0.01 | **35.19 ± 0.34** | OOM | 45.77 ± 0.22 | 44.95 ± 0.13 | 38.23 ± 0.29 | 36.70 ± 0.29 |
| KN | MAE | OOM | 52.98 ± 1.36 | 116.61 ± 0.23 | **42.78 ± 0.05** | 44.75 ± 0.24 | 49.44 ± 0.46 | 57.22 ± 1.96 |
| | RMSE | OOM | 83.22 ± 0.84 | 149.74 ± 0.66 | 67.32 ± 0.14 | **65.83 ± 0.35** | 79.03 ± 0.50 | 86.98 ± 2.16 |
| | MAPE(%) | OOM | **54.05 ± 3.83** | 297.19 ± 1.49 | 70.57 ± 0.27 | 77.29 ± 3.06 | 61.92 ± 1.39 | 89.88 ± 6.97 |
| DA | MAE | 56.88 ± 0.01 | **56.19 ± 0.15** | OOM | 59.55 ± 0.01 | 61.34 ± 0.24 | 57.52 ± 0.40 | 57.85 ± 0.09 |
| | RMSE | 88.16 ± 0.02 | **86.45 ± 0.70** | OOM | 90.81 ± 0.06 | 90.98 ± 0.34 | 87.83 ± 1.01 | 90.13 ± 0.16 |
| | MAPE(%) | 53.29 ± 0.09 | 54.39 ± 6.00 | OOM | 57.20 ± 0.02 | 66.56 ± 0.19 | 55.03 ± 0.35 | **52.07 ± 0.04** |
| ESS | MAE | 44.08 ± 0.27 | 44.91 ± 0.32 | 179.03 ± 1.10 | 62.74 ± 0.04 | 49.97 ± 0.14 | 41.68 ± 0.05 | **41.42 ± 0.27** |
| | RMSE | 65.43 ± 0.68 | 67.98 ± 0.93 | 232.52 ± 0.26 | 92.85 ± 0.16 | 73.19 ± 0.40 | **62.40 ± 0.24** | 63.04 ± 0.50 |
| | MAPE(%) | **35.95 ± 1.15** | 49.74 ± 5.81 | 308.94 ± 8.39 | 55.33 ± 1.11 | 48.59 ± 1.21 | 37.57 ± 0.39 | 38.41 ± 2.97 |
| FRA | MAE | 205.72 ± 0.83 | 205.31 ± 22.15 | 168.51 ± 1.59 | **109.03 ± 36.74** | 132.32 ± 1.38 | 250.59 ± 11.89 | 386.51 ± 26.96 |
| | RMSE | 238.54 ± 0.79 | 234.38 ± 24.27 | 209.58 ± 0.03 | **132.51 ± 38.23** | 165.66 ± 0.15 | 282.51 ± 11.51 | 408.51 ± 27.19 |
| | MAPE(%) | 73.26 ± 0.25 | 70.54 ± 5.11 | 55.50 ± 0.97 | **35.27 ± 11.29** | 41.22 ± 0.39 | 85.66 ± 5.06 | 132.85 ± 10.28 |
| GRZ | MAE | 63.46 ± 0.17 | **57.03 ± 0.76** | 191.54 ± 0.00 | 71.93 ± 0.01 | 63.14 ± 0.03 | 62.37 ± 0.78 | 59.51 ± 0.28 |
| | RMSE | 95.67 ± 0.07 | **83.99 ± 0.61** | 241.77 ± 0.00 | 103.75 ± 0.01 | 91.38 ± 0.00 | 92.58 ± 0.82 | 95.09 ± 0.61 |
| | MAPE(%) | 109.30 ± 3.51 | 77.93 ± 5.43 | 460.74 ± 0.00 | 81.48 ± 0.13 | 78.44 ± 0.27 | **70.16 ± 4.58** | 77.40 ± 2.29 |
| GRQ | MAE | 77.05 ± 0.27 | **72.08 ± 3.04** | 150.91 ± 0.00 | 72.34 ± 1.76 | 83.37 ± 2.14 | 72.33 ± 0.77 | 82.93 ± 6.04 |
| | RMSE | 104.16 ± 0.05 | **98.38 ± 2.63** | 205.49 ± 0.00 | 100.03 ± 0.44 | 115.72 ± 2.69 | 100.25 ± 0.83 | 110.28 ± 7.48 |
| | MAPE(%) | 41.63 ± 0.69 | **35.60 ± 1.85** | 117.24 ± 0.00 | 44.18 ± 2.38 | 46.94 ± 1.33 | 38.69 ± 0.88 | 45.80 ± 2.96 |
| HAM | MAE | 47.08 ± 0.23 | 45.88 ± 0.41 | 97.51 ± 0.59 | 48.97 ± 0.02 | 49.77 ± 0.01 | **45.35 ± 0.05** | 46.06 ± 0.01 |
| | RMSE | 80.67 ± 0.53 | 78.49 ± 1.08 | 150.82 ± 2.57 | 84.20 ± 0.08 | 83.90 ± 0.06 | **77.70 ± 0.01** | 82.77 ± 2.08 |
| | MAPE(%) | 45.78 ± 0.83 | **44.31 ± 3.07** | 111.41 ± 4.58 | 52.32 ± 0.20 | 53.06 ± 0.21 | 44.81 ± 0.31 | 45.29 ± 0.06 |
| INN | MAE | 75.51 ± 0.80 | 72.05 ± 0.00 | 347.11 ± 2.98 | 106.94 ± 0.00 | 80.50 ± 1.00 | **69.08 ± 0.43** | OOM |
| | RMSE | 110.87 ± 0.65 | 106.99 ± 0.06 | 464.48 ± 8.39 | 173.49 ± 0.15 | 117.77 ± 1.61 | **100.59 ± 0.79** | OOM |
| | MAPE(%) | 31.27 ± 1.22 | 36.38 ± 0.72 | 300.48 ± 37.49 | 46.75 ± 0.75 | 34.23 ± 0.15 | **27.74 ± 0.06** | OOM |
| KS | MAE | 97.97 ± 0.72 | 103.19 ± 5.03 | 227.96 ± 0.00 | **78.86 ± 14.60** | 104.59 ± 1.19 | 80.04 ± 0.72 | 224.87 ± 1.62 |
| | RMSE | 244.74 ± 0.48 | 244.15 ± 6.33 | 332.77 ± 0.00 | **188.64 ± 2.13** | 209.20 ± 1.15 | 195.69 ± 0.26 | 314.80 ± 4.13 |
| | MAPE(%) | 143.62 ± 1.84 | 156.53 ± 13.18 | 433.64 ± 0.00 | 110.39 ± 41.23 | 162.29 ± 2.84 | **109.72 ± 2.55** | 407.04 ± 1.53 |

| | | | | | | | | |
|---|---|---|---|---|---|---|---|---|
| MAN | MAE | $117.20 \pm 2.55$ | $109.56 \pm 2.55$ | $336.84 \pm 0.00$ | $\mathbf{103.86 \pm 9.84}$ | $123.20 \pm 0.27$ | $107.88 \pm 1.14$ | $107.97 \pm 2.60$ |
| | RMSE | $196.04 \pm 4.11$ | $190.75 \pm 4.72$ | $450.82 \pm 0.00$ | $\mathbf{173.23 \pm 12.91}$ | $198.08 \pm 0.51$ | $181.98 \pm 4.00$ | $186.65 \pm 3.39$ |
| | MAPE(%) | $47.38 \pm 0.86$ | $45.64 \pm 2.89$ | $278.97 \pm 0.00$ | $46.64 \pm 3.55$ | $60.39 \pm 0.14$ | $\mathbf{44.84 \pm 1.22}$ | $47.02 \pm 1.75$ |
| MEL | MAE | $64.79 \pm 0.03$ | $\mathbf{54.63 \pm 4.48}$ | OOM | $87.95 \pm 1.60$ | $67.32 \pm 0.31$ | $69.54 \pm 1.08$ | $56.73 \pm 0.94$ |
| | RMSE | $101.09 \pm 0.06$ | $\mathbf{81.23 \pm 6.52}$ | OOM | $127.60 \pm 1.41$ | $97.39 \pm 0.04$ | $105.83 \pm 1.53$ | $87.59 \pm 0.91$ |
| | MAPE(%) | $55.34 \pm 0.02$ | $49.10 \pm 5.14$ | OOM | $103.87 \pm 4.79$ | $70.74 \pm 1.14$ | $\mathbf{47.17 \pm 0.20}$ | $50.21 \pm 0.12$ |
| RTM | MAE | $\mathbf{57.11 \pm 0.15}$ | $58.08 \pm 1.24$ | $189.90 \pm 0.00$ | $85.30 \pm 0.50$ | $82.23 \pm 0.36$ | $85.20 \pm 1.78$ | $65.01 \pm 0.31$ |
| | RMSE | $99.90 \pm 0.34$ | $\mathbf{98.98 \pm 0.53}$ | $250.70 \pm 0.00$ | $141.23 \pm 0.46$ | $128.17 \pm 0.09$ | $136.93 \pm 2.26$ | $110.64 \pm 0.98$ |
| | MAPE(%) | $\mathbf{43.47 \pm 0.63}$ | $55.83 \pm 11.21$ | $373.03 \pm 0.00$ | $65.00 \pm 0.87$ | $76.28 \pm 0.17$ | $62.30 \pm 2.15$ | $46.88 \pm 1.28$ |
| SDR | MAE | $121.60 \pm 0.24$ | $129.77 \pm 6.53$ | $272.16 \pm 0.00$ | $120.00 \pm 0.03$ | $152.27 \pm 0.77$ | $\mathbf{104.78 \pm 0.34}$ | $119.57 \pm 0.47$ |
| | RMSE | $278.70 \pm 0.41$ | $279.99 \pm 1.64$ | $440.08 \pm 0.00$ | $262.91 \pm 0.01$ | $284.60 \pm 0.05$ | $\mathbf{249.10 \pm 0.76}$ | $274.87 \pm 3.75$ |
| | MAPE(%) | $67.07 \pm 2.54$ | $92.03 \pm 14.14$ | $301.72 \pm 0.00$ | $68.81 \pm 0.28$ | $119.83 \pm 1.98$ | $58.89 \pm 5.49$ | $\mathbf{55.51 \pm 0.36}$ |
| SP | MAE | $49.26 \pm 0.11$ | $\mathbf{48.39 \pm 0.19}$ | $122.84 \pm 4.61$ | $55.70 \pm 0.01$ | $54.59 \pm 0.03$ | $49.12 \pm 0.17$ | $48.70 \pm 0.25$ |
| | RMSE | $70.97 \pm 0.13$ | $\mathbf{69.48 \pm 0.62}$ | $175.83 \pm 13.31$ | $79.94 \pm 0.01$ | $77.28 \pm 0.03$ | $70.95 \pm 0.34$ | $70.28 \pm 0.31$ |
| | MAPE(%) | $39.90 \pm 0.46$ | $38.25 \pm 1.95$ | $102.22 \pm 24.13$ | $47.80 \pm 0.10$ | $46.68 \pm 0.32$ | $38.51 \pm 0.41$ | $\mathbf{37.52 \pm 0.37}$ |
| SXB | MAE | $80.33 \pm 0.24$ | $\mathbf{78.32 \pm 0.23}$ | $262.40 \pm 0.00$ | $94.09 \pm 0.05$ | $89.88 \pm 0.17$ | $80.19 \pm 0.36$ | $78.71 \pm 0.29$ |
| | RMSE | $142.42 \pm 0.73$ | $\mathbf{139.91 \pm 0.95}$ | $363.49 \pm 0.00$ | $162.33 \pm 0.05$ | $156.75 \pm 0.32$ | $141.91 \pm 0.63$ | $141.32 \pm 0.36$ |
| | MAPE(%) | $40.29 \pm 0.54$ | $38.22 \pm 0.97$ | $223.27 \pm 0.00$ | $51.18 \pm 0.37$ | $49.09 \pm 0.17$ | $40.96 \pm 0.76$ | $\mathbf{37.89 \pm 0.11}$ |
| STR | MAE | $61.10 \pm 0.38$ | $57.78 \pm 0.84$ | $71.86 \pm 5.01$ | $73.59 \pm 0.94$ | $68.30 \pm 0.33$ | $\mathbf{56.40 \pm 0.08}$ | OOM |
| | RMSE | $81.05 \pm 0.55$ | $74.94 \pm 1.18$ | $96.17 \pm 7.97$ | $96.18 \pm 1.48$ | $90.50 \pm 0.70$ | $\mathbf{73.46 \pm 0.06}$ | OOM |
| | MAPE(%) | $22.32 \pm 0.19$ | $21.28 \pm 1.87$ | $25.56 \pm 2.23$ | $30.23 \pm 0.66$ | $25.02 \pm 1.17$ | $\mathbf{20.36 \pm 0.09}$ | OOM |
| TPE | MAE | $147.54 \pm 0.61$ | $\mathbf{140.98 \pm 2.89}$ | $499.95 \pm 8.83$ | $163.96 \pm 0.02$ | $168.31 \pm 0.17$ | $142.71 \pm 0.55$ | $142.86 \pm 1.83$ |
| | RMSE | $670.14 \pm 0.45$ | $671.29 \pm 2.58$ | $996.04 \pm 7.32$ | $726.80 \pm 0.40$ | $726.73 \pm 1.06$ | $\mathbf{666.42 \pm 0.36}$ | $678.02 \pm 3.41$ |
| | MAPE(%) | $54.90 \pm 0.81$ | $47.50 \pm 4.82$ | $280.63 \pm 13.88$ | $52.22 \pm 0.17$ | $60.40 \pm 0.77$ | $45.94 \pm 2.11$ | $\mathbf{44.89 \pm 0.61}$ |
| TO | MAE | $104.21 \pm 0.69$ | $\mathbf{92.97 \pm 2.72}$ | $319.07 \pm 5.97$ | $106.30 \pm 0.02$ | $105.57 \pm 0.66$ | $139.93 \pm 0.61$ | $133.21 \pm 1.02$ |
| | RMSE | $178.09 \pm 1.38$ | $\mathbf{154.59 \pm 2.71}$ | $424.02 \pm 8.74$ | $182.49 \pm 0.32$ | $167.92 \pm 0.92$ | $231.65 \pm 2.32$ | $221.49 \pm 0.33$ |
| | MAPE(%) | $67.26 \pm 0.52$ | $\mathbf{51.82 \pm 1.26}$ | $400.37 \pm 6.15$ | $59.46 \pm 1.38$ | $69.03 \pm 0.16$ | $82.81 \pm 0.41$ | $87.58 \pm 0.12$ |
| YTO | MAE | $57.16 \pm 0.24$ | $\mathbf{56.66 \pm 1.67}$ | $173.86 \pm 9.18$ | $125.57 \pm 0.08$ | $74.96 \pm 1.04$ | $70.85 \pm 0.31$ | $60.80 \pm 0.21$ |
| | RMSE | $94.09 \pm 0.11$ | $\mathbf{90.75 \pm 1.41}$ | $253.42 \pm 9.38$ | $191.73 \pm 0.25$ | $114.34 \pm 1.97$ | $115.62 \pm 0.60$ | $100.24 \pm 0.45$ |
| | MAPE(%) | $44.91 \pm 1.53$ | $\mathbf{43.39 \pm 4.12}$ | $128.97 \pm 45.99$ | $110.73 \pm 1.97$ | $73.18 \pm 1.56$ | $44.39 \pm 0.43$ | $43.66 \pm 0.17$ |
| TLS | MAE | $257.57 \pm 0.47$ | $255.17 \pm 0.18$ | $264.44 \pm 6.13$ | $264.06 \pm 0.00$ | $298.73 \pm 0.16$ | $\mathbf{255.11 \pm 0.01}$ | $257.76 \pm 0.30$ |
| | RMSE | $348.95 \pm 1.83$ | $342.20 \pm 1.02$ | $352.99 \pm 9.76$ | $348.31 \pm 0.02$ | $408.17 \pm 0.23$ | $\mathbf{340.28 \pm 0.14}$ | $346.83 \pm 0.97$ |
| | MAPE(%) | $746.15 \pm 8.05$ | $758.84 \pm 36.31$ | $791.75 \pm 78.81$ | $867.12 \pm 0.77$ | $842.35 \pm 8.43$ | $\mathbf{745.07 \pm 5.77}$ | $746.42 \pm 0.82$ |
| UTC | MAE | OOM | $55.13 \pm 4.47$ | OOM | $57.76 \pm 0.72$ | $73.45 \pm 1.74$ | $73.81 \pm 0.57$ | $\mathbf{40.36 \pm 0.11}$ |
| | RMSE | OOM | $89.55 \pm 1.67$ | OOM | $97.52 \pm 1.24$ | $109.98 \pm 1.82$ | $123.95 \pm 2.23$ | $\mathbf{75.86 \pm 0.31}$ |
| | MAPE(%) | OOM | $80.70 \pm 18.35$ | OOM | $64.93 \pm 0.77$ | $105.19 \pm 2.55$ | $93.66 \pm 5.16$ | $\mathbf{40.85 \pm 2.82}$ |
| VNO | MAE | $96.16 \pm 0.15$ | $91.31 \pm 0.19$ | OOM | $81.71 \pm 4.58$ | $98.47 \pm 0.10$ | $\mathbf{80.01 \pm 0.59}$ | $105.63 \pm 1.55$ |
| | RMSE | $130.79 \pm 0.34$ | $124.11 \pm 0.44$ | OOM | $111.45 \pm 2.76$ | $131.28 \pm 0.02$ | $\mathbf{109.93 \pm 0.18}$ | $141.02 \pm 1.91$ |
| | MAPE(%) | $63.46 \pm 0.21$ | $58.74 \pm 1.31$ | OOM | $47.91 \pm 8.96$ | $60.49 \pm 0.08$ | $\mathbf{46.17 \pm 1.50}$ | $77.00 \pm 2.31$ |
| WOB | MAE | $56.73 \pm 0.26$ | $\mathbf{53.73 \pm 0.93}$ | $65.57 \pm 0.76$ | $69.50 \pm 0.05$ | $61.35 \pm 0.11$ | $57.82 \pm 0.18$ | $56.30 \pm 0.43$ |
| | RMSE | $90.97 \pm 0.48$ | $\mathbf{83.48 \pm 2.09}$ | $104.61 \pm 1.31$ | $111.82 \pm 0.01$ | $92.81 \pm 0.13$ | $91.64 \pm 0.05$ | $89.31 \pm 0.69$ |
| | MAPE(%) | $42.48 \pm 1.04$ | $\mathbf{41.16 \pm 3.01}$ | $53.82 \pm 1.86$ | $55.70 \pm 0.59$ | $54.53 \pm 0.12$ | $44.75 \pm 0.61$ | $43.14 \pm 1.57$ |
| ZRH | MAE | OOM | $56.73 \pm 0.03$ | OOM | $65.62 \pm 0.00$ | $65.29 \pm 0.20$ | $61.77 \pm 4.83$ | $\mathbf{54.71 \pm 0.20}$ |
| | RMSE | OOM | $80.80 \pm 0.23$ | OOM | $94.06 \pm 0.01$ | $91.54 \pm 0.20$ | $88.03 \pm 6.88$ | $\mathbf{77.75 \pm 0.44}$ |
| | MAPE(%) | OOM | $36.22 \pm 0.43$ | OOM | $47.73 \pm 0.13$ | $48.20 \pm 0.21$ | $46.66 \pm 8.57$ | $\mathbf{35.87 \pm 0.06}$ |

# E    LIMITATIONS

- **Data quality needs more researchers to verify.** Due to the fact that OpenStreetMap is a free, open geographic database updated and maintained by a community of volunteers, the raw data from OSM has not been thoroughly validated. Thus, we need more researchers from academia and industries to join us to improve the data quality.

- **Application-unique APIs need more researchers to develop.** Although we have released the basic APIs for querying and processing the data which can be used for massive applications, each application may require further unique processing steps. Thus, we kindly encourage more researchers to contribute to the OSM+ dataset, processing tools and downstream applications.

# F   AUTHOR STATEMENT

We want to show our great thanks to OpenStreetMap (Haklay & Weber, 2008a), UTD19 (Loder et al., 2020), Oak Ridge National Laboratory (Sims et al., 2023) and NASA Earth Observatory (Weier, 2003) for providing free-public roadnet data, traffic flow data, population data and nightlight data. Please let us know if any issues are found. We will take appropriate action when needed, e.g. to remove data records with such issues.

