# OpenReview forum: "OSM+: Cloud-native Open Street Map Data System for City-wide Experiments"
_ICLR.cc/2025/Conference — ICLR 2025 Conference Withdrawn Submission_

### Official Review · Reviewer_JMUY · 2024-10-27

**Soundness:** 2
**Presentation:** 2
**Contribution:** 1
**Rating:** 3
**Confidence:** 3

**Summary:**

The paper organizes OpenStreetMap and presents a structured global network graph database containing a billion nodes, for better accessibility and availability. The usefulness is demonstrated in three cases, traffic prediction task, city boundary detection task and traffic policy control task. For the extensively studied traffic forecasting task, it has introduced a new benchmark featuring 31 datasets. Overall, the core could be useful for spatial databasing and OSM downstream analysis, but it may be out of the scope of ICLR.

**Strengths:**

1. This paper develops a road network graph database using OpenStreetMap, aiming to address the issues that processed data is often not well-defined or uniformly structured and is typically only used for one-time applications.
2. Based on the proposed road database, a series of basic computing APIs are introduced to allow efficient node query and distance query.
3. Customize road network data for typical urban research questions to facilitate experiments on datasets.

**Weaknesses:**

1. Better data organization could be useful, but the paper lacks novelty, and does not fit ICLR.
2. The paper mentions the sensors and the sparsity challenge. However, the paper does not provide information on the source of the sensor data, and it is not reflected in the three  tables of the databases discussed. Further, the paper lacks a sufficient explanation of how the Sparsity Challenge relates to the advantages of the database .
3. Figure 6 in the paper does not seem to be referenced or explained.
4. In the experiment of traffic prediction, the OSM database uses the UTD19 dataset and is compared with the PEMS dataset. But it appears that the advantages of the OSM database are not evident from this comparison. The variance problem probably comes more from UTD19 than from the OSM database.

**Questions:**

1. Explain the question in weakness about sensors and the sparsity challenge.
2. Figure 6 in the paper does not seem to be referenced or explained.What does Figure 6 aim to convey?
3. Explain the question in weakness about experiment of traffic prediction.

---

### Official Review · Reviewer_WTpi · 2024-10-31

**Soundness:** 2
**Presentation:** 3
**Contribution:** 2
**Rating:** 3
**Confidence:** 4

**Summary:**

This research presents OSM+, a large-scale road network computing engine built on OpenStreetMap to support urban research applications. OSM+ includes efficient APIs for spatial queries and data converters, facilitating tasks such as traffic prediction, policy control, and boundary detection. The paper demonstrates significant engineering and data processing efforts. However, the contributions and validation in this version of the paper are not clearly articulated, and revisions may be needed in the areas mentioned in the weaknesses and questions sections.

Below are my initial comments, and I will consider revising my scores if the authors provide additional clarifications and address the identified weaknesses during the discussion period.

**Strengths:**

This research showcases a large-scale system that effectively integrates data from multiple sources, demonstrating substantial engineering efforts. I recognize the authors' efforts to build a foundation for urban and AI research.

**Weaknesses:**

1. The paper uses varying terms like system, framework, and engine to describe the proposed work, which makes it confusing for me to identify differences between them.

2. The frequent use of the term "efficient" throughout the text lacks evidence (i.e., experiments) to support it.

3. Figure 4 is not referenced in the main text.

4. Some logical flow issues arise, such as in Section 4.1: "However, in recent years, it has been noticed that newly proposed methods can hardly exhibit significant improvements over existing ones. Thus, we are here to propose 31 new city-level datasets." Introducing new datasets does not necessarily address marginal improvements in existing methods. Additionally, the main motivation for proposing a large number of datasets seems to be to highlight their dynamic and sparse characteristics. However, there is no statistical evidence demonstrating how sparsity varies between cities compared to classic datasets like PEMS and METR.

5. The spatial and temporal spans of the 31 traffic flow datasets are not clearly outlined, nor is the process for data collection (e.g., traffic flow data) explained.

6. The traffic policy control experiment lacks clarity, and no results for this experiment are presented in the main text.

7. Overall, the experiments section offers limited insights that might benefit future research, aside from the city boundary detection task. The findings in the city boundary detection task seem only loosely related to the system’s contributions. It's unclear how this system differs from existing ones in identifying correlations between road networks and socio-economic indicators.

8. Each contribution mentioned by the authors is not sufficiently addressed:
- Data-wise: How is this data different from typical OSM data? If efficiency is a key feature of the proposed system, efficiency experiments should be included in the main experiments.
- System-wise: The system’s uniqueness compared to other systems is not clearly articulated.
- Application-wise: The definition and scope of applications, such as traffic signal control, are not clearly explained.
- Benchmark-wise: The motivation behind the benchmark and the relevant dataset statistics are missing.

**Questions:**

1. What are the differences between the terms "system", "framework", and "engine" as used in this paper?
2. Is "window check" a technical contribution of this paper? From my experience, similar techniques like spatial indexing exist.
3. Could the authors provide more details on the contribution to storing road networks as graph-structured data? My understanding is that such graph data could be obtained relatively easily using the OSM API and packages like NetworkX. Are there any specific procedures designed for processing this graph data? For example, how many subgraphs are there? Did the authors merge different subgraphs or make other efforts to refine this data?
4. Could the authors clarify the traffic policy control experiment?

---

### Official Review · Reviewer_CfRR · 2024-11-03

**Soundness:** 3
**Presentation:** 3
**Contribution:** 2
**Rating:** 3
**Confidence:** 5

**Summary:**

This paper introduces OSM+, a global road network map database designed to support city-wide research.OSM+ solves the problems of computational resources and format uniformity required for processing large-scale road data in urban research by building a structured database from OpenStreetMap data.OSM+ contains three main components: a global-scale road network map database, a series of automated parallel computing APIs that support efficient node and distance queries, and a data transformation tool for urban research. The paper also describes three application scenarios: a traffic prediction task, a transportation policy control task, and a city boundary detection task. The system reduces data preprocessing time, facilitates the fusion of multimodal spatial-temporal data, supports large model training, and accelerates the process of scientific discovery.

**Strengths:**

1. This paper reconstructs a worldwide structured road network database based on OpenStreetMap, which is a valuable resource for urban research and can significantly improve the ease and efficiency of related research.
2. The OSM+ proposed in this paper is able to support a variety of urban research tasks, including traffic prediction, traffic control, and urban boundary detection, which provides opportunities for exploration and experimentation in related areas of research.
3. The databases and tools provided in this paper help to enable efficient querying and data processing, support large-scale urban study, and reduce the time and computational resource overhead for road network data processing.
4. This paper supports the use of map data for the fusion of multimodal spatio-temporal data, suitable for modern training of more complex large models and realizing innovative scientific discoveries.

**Weaknesses:**

1. Although the paper mentions a series of data processing steps, the lack of a detailed description of the data cleaning and transformation process puts the credibility of the data quality into question, and furthermore, the data for the paper appears to be directly imported and then converted to its corresponding data directly through osm2pqsql. https://planet.openstreetmap.org/ .
2. Although OSM+ provides standardized global road network data, this is based on existing OpenStreetMap data and fails to clearly articulate its distinctive and innovative contribution
3. While this paper cites applications such as traffic forecasting, transportation policy control, and urban boundary detection, the examples are not sufficiently in-depth to adequately demonstrate the specific ways in which OSM+ significantly outperforms other methods in these applications.
4. This paper does not provide quantitative comparisons to demonstrate the performance benefits of OSM+ over other existing tools and formats.

**Questions:**

Existing OSMs are able to provide corresponding query api, which may vary due to different network conditions. However, this paper does not fully explain the advantages and performance improvement of OSM+ over existing open source or commercial map data tools (e.g., OSMnx, etc.), which makes it difficult for readers to understand its relative innovations and improvements clearly.

**Details Of Ethics Concerns:**

The data in this paper is basically derived from OpenStreetMap's database, but the authors did not follow the organization's rules on data copyright.Please see https://www.openstreetmap.org/copyright

---

### Official Review · Reviewer_srQs · 2024-11-03

**Soundness:** 2
**Presentation:** 2
**Contribution:** 2
**Rating:** 5
**Confidence:** 4

**Summary:**

This paper introduces a framework for processing large-scale road network data from OpenStreetMap, called OSM+. The framework consists of three components: a road network graph database, computing APIs, and a suite of data converters. Utilizing the processed road network data from OSM+, the authors present three illustrative use cases—traffic prediction, city boundary detection, and traffic policy control—to showcase the framework's usability.

**Strengths:**

1. OSM+ provides open-source, large-scale road network data, offering an easy-to-use resource for developing various algorithms and advancing research within the related community.
2. The diverse use cases highlight the usability of OSM+, with particular effectiveness demonstrated in traffic prediction tasks.

**Weaknesses:**

1. The technical contribution of the paper is unclear. From the current description of the graph database and query APIs, the advancements introduced by OSM+ appear limited. Although some improvements are mentioned in Section 3, what specific technical methods does OSM+ propose to achieve these advancements?

2. The first use case on traffic prediction is well-explained; however, more comparative results with other large-scale traffic datasets, such as [1], should be provided to better demonstrate the value of OSM+ for traffic prediction tasks.

[1] Liu, Xu, et al. "Largest: A Benchmark Dataset for Large-Scale Traffic Forecasting." Advances in Neural Information Processing Systems 36 (2024).

3. The results for the other two use cases are overly simplistic. More detailed methods and comprehensive results are needed to demonstrate the effectiveness of OSM+ in these applications. Additionally, it is unclear how OSM+ contributes to research in these two areas. For instance, could the authors provide results that demonstrate the types of analyses or experiments made possible by OSM+?

**Questions:**

1. Additional comparative results with other large-scale traffic datasets, such as [1], would be beneficial. For example, how does OSM+ offer new insights compared to LargeST in the context of traffic prediction tasks?

[1] Liu, Xu, et al. "Largest: A Benchmark Dataset for Large-Scale Traffic Forecasting." Advances in Neural Information Processing Systems 36 (2024).

2. More detailed methods and comprehensive results on the last two use cases. For instance, could the authors provide results that demonstrate the types of analyses or experiments made possible by OSM+?

---

### Note · Authors · 2024-11-21

I have read and agree with the venue's withdrawal policy on behalf of myself and my co-authors.